# Deficiency in DNAH12 causes male infertility by impairing DNAH1 and DNALI1 recruitment in humans and mice

Menglei Yang, Hafiz Muhammad Jafar Hussain, Manan Khan, Zubair Muhammad, Jianteng Zhou, Ao Ma, Xiongheng Huang, Jingwei Ye, Min Chen, Aoran Zhi, Tao Liu, Ranjha Khan, Ali Asim, Wasim Shah, Aurang Zeb, Nisar Ahmad, Huan Zhang, Bo Xu, Hui Ma*, Qinghua Shi*, Baolu Shi*

Center for Reproduction and Genetics, Department of Obstetrics and Gynecology, First Affiliated Hospital of USTC, Hefei National Research Center for Physical Sciences at the Microscale, the CAS Key Laboratory of Innate Immunity and Chronic Disease, School of Basic Medical Sciences, Biomedical Sciences and Health Laboratory of Anhui Province, Institute of Health and Medicine, Hefei Comprehensive National Science Center, Division of Life Sciences and Medicine, University of Science and Technology of China, Hefei, China

*For correspondence:
clsmh@ustc.edu.cn (HM);
qshi@ustc.edu.cn (QS);
bl625@ustc.edu.cn (BS)

## eLife Assessment

This **fundamental** study further validates DNAH12 as a causative gene for asthenoteratozoospermia and male infertility in both humans and mice. **Compelling** evidence supports the notion that DNAH12 is essential for proper axonemal development. This work will be of interest to reproductive biologists studying spermatogenesis and sperm biology, as well as andrologists focusing on male fertility.

**Abstract** Asthenoteratozoospermia, a prevalent cause of male infertility, lacks a well-defined etiology. DNAH12 is a special dynein featured by the absence of a microtubule-binding domain, however, its functions in spermatogenesis remain largely unknown. Through comprehensive genetic analyses involving whole-exome sequencing and subsequent Sanger sequencing on infertile patients and fertile controls from six distinct families, we unveiled six biallelic mutations in *DNAH12* that co-segregate recessively with male infertility in the studied families. Transmission electron microscopy (TEM) revealed pronounced axonemal abnormalities, including inner dynein arms (IDAs) impairment and central pair (CP) loss in sperm flagella of the patients. Mouse models (*Dnah12*-/- and *Dnah12*mut/mut) were generated and recapitulated the reproductive defects in the patients. Noteworthy, DNAH12 deficiency did not show effects on cilium organization and function. Mechanistically, DNAH12 was confirmed to interact with two other IDA components DNALI1 and DNAH1, while disruption of DNAH12 leads to failed recruitment of DNALI1 and DNAH1 to IDAs and compromised sperm development. Furthermore, DNAH12 also interacts with radial spoke head proteins RSPH1, RSPH9, and DNAJB13 to regulate CP stability. Moreover, the infertility of *Dnah12*-/- mice could be overcome by intracytoplasmic sperm injection (ICSI) treatment. Collectively, DNAH12 plays a crucial role in the proper organization of axoneme in sperm flagella, but not cilia, by recruiting DNAH1 and DNALI1 in both humans and mice. These findings expand our comprehension of dynein component assembly in flagella and cilia and provide a valuable marker for genetic counseling and diagnosis of asthenoteratozoospermia in clinical practice.

## Introduction

Infertility has emerged as a global health concern, impacting around 8–12% of couples during their reproductive years, approximately 50% of infertility cases are attributed to male factors (*Agarwal et al., 2021*). Among these, asthenoteratozoospermia, characterized by poor sperm motility and notable morphological abnormalities, stands out as a prevalent contributor to male infertility (*Krausz and Riera-Escamilla, 2018*; *Shang et al., 2018*). Recent investigations have unveiled a multitude of gene mutations associated with asthenoteratozoospermia and male infertility, including Dynein Axonemal Heavy Chain (DNAH) inner dynein arm (IDA) genes like *DNAH1*, *DNAH2*, *DNAH6*, *DNAH7*, and *DNAH10*, as well as outer dynein arm (ODA) genes such as *DNAH8* and *DNAH17*, all implicated in primary male infertility with asthenoteratozoospermia (*Khan et al., 2021*; *Li et al., 2019*; *Tu et al., 2019*; *Gao et al., 2022a*; *Li et al., 2022*; *Liu et al., 2020*; *Whitfield et al., 2019*; *Zhang et al., 2020*). Additionally, genes encoding cilia and flagella-associated proteins (CFAP), including *CFAP43*, *CFAP44*, *CFAP57*, and *CFAP61*, have been identified as linked to asthenoteratozoospermia (*Coutton et al., 2018*; *Tang et al., 2017*; *Ma et al., 2023*; *Hu et al., 2023a*). Despite these advances, the intricate and heterogeneous features of asthenoteratozoospermia leave a significant number of cases unexplained. It is anticipated that additional genetic factors play a role in sperm anomalies and dysfunction.

The axoneme, a fundamental structure of cilia and flagella, exhibits a well-organized pattern with nine peripheral doublet microtubules (DMTs) and a central pair of microtubules (CP), constituting the classical '9+2' microtubular arrangement (*Touré et al., 2021*). In motile cilia or flagella, the radial spoke (RS) head contacts the CP accessory structures, transmitting mechanical and chemical signals from the central microtubules to the axonemal dynein arms, thus coordinating the flagellar or ciliary movement (*Walton et al., 2023*). Within mammalian species, the DNAH family plays a crucial role in the axonemal structure, encompassing eight IDA members (DNAH1-3, DNAH6, DNAH7, DNAH10, DNAH12, DNAH14) and five ODA members (DNAH5, DNAH8, DNAH9, DNAH11, DNAH17). Despite the collaborative effort of IDAs and ODAs in generating the force necessary for flagellar and ciliary motion, they serve distinct functions. Chlamydomonas ODAs, composed of α, β, and γ chains, primarily regulate beat frequency, while IDAs, with seven major subspecies (a-g), contribute to bending formation and beating form (*Viswanadha et al., 2017*). Mutations in genes encoding ODA or IDA components have been linked to asthenoteratozoospermia, with or without symptoms of primary ciliary dyskinesia (PCD), a genetic disorder characterized by chronic airway infections or situs inversus (*Kleinboelting et al., 2014*). Notably, deficiency in certain IDAs and ODAs coding genes, such as *DNAH7* or *DNAH11*, may lead to PCD (*Shoemark et al., 2018*; *Sironen et al., 2020*; *Wei et al., 2021*). Patients carrying mutations in *DNAH1* and *DNAH2* typically lack PCD symptoms (*Khan et al., 2021*), although a study suggested the potential contribution of the DNAH1 p.Lys1154Gln mutation to PCD (*Imtiaz et al., 2015*). The extensive genetic heterogeneity and clinical variabilities underscore the intricacy and uniqueness of dynein arms in flagella and cilia. However, the susceptibility of uncharacterized dynein factors that are related to asthenoteratozoospermia and/or PCD remains unclear, given the absence of human genetic evidence or corresponding animal models.

Dynein axonemal heavy chain 12, DNAH12 (MIM: 603340, GenBank: NM_001366028.2), also known as DHC3, is a unique dynein arm component (*Vaisberg et al., 1996*). Notably, it is the shortest member and lacks a typical microtubule-binding domain (MTBD) in the DNAH family. Mutations of *DNAH12* have been predicted to be associated with male patients' infertility, however, limited pathogenic evidence was provided, and the underlying pathogenesis remained unclear (*Li et al., 2021*; *Oud et al., 2021*). Besides, although DNAH12 has been identified as a possible axonemal dynein and its mRNA is predominantly detected in two tissues containing axonemal structures: trachea and testis (*Vaisberg et al., 1996*; *Chapelin et al., 1997*), its physiological functions remain largely elusive.

In this study, we enrolled 796 infertile patients from unrelated Pakistani and Chinese families diagnosed with asthenozoospermia. Utilizing whole-exome sequencing (WES) followed by Sanger sequencing, we identified six novel bi-allelic mutations in *DNAH12* in infertile males from six unrelated families. All these mutations resulted in the loss of DNAH12 protein in the sperm of patients. Two mouse models, *Dnah12⁻/⁻* and *Dnah12^{mut/mut}*, were generated and exhibited asthenoteratozoospermia and male infertility, mimicking our patients' phenotypes. Deficiency in DNAH12 led to axoneme defects characterized by impaired IDAs and CP loss in sperm flagella, while no such defects were observed in cilia. DNAH12 interacts with other IDA components like DNALI1 and DNAH1 in

mouse testes, and disruption of DNAH12 impaired the recruitment of dynein components into sperm flagella. Our findings unveil, for the first time, that DNAH12 plays a pivotal role in facilitating the recruitment of IDA components specifically in flagella, but not in cilia. Furthermore, we demonstrate that bi-allelic mutations in *DNAH12* are identified as causative factors for asthenoteratozoospermia and male infertility in both humans and mice. These results enhance our understanding of the development of axonemal structures in flagella versus cilia and provide valuable clues for genetic counseling and future individualized treatment for infertile males carrying *DNAH12* mutations in clinical practice.

## Results

### Identification of biallelic *DNAH12* variants in men with asthenoteratozoospermia

To identify the genetic factors contributing to male infertility, we recruited patients with primary infertility and their family members from Pakistan and China. WES was performed on these individuals, followed by comprehensive analyses of the WES data (*Figure 1—figure supplement 1A–1C*). Seven mutations in *DNAH12* were identified as potential pathogenic factors for male infertility in patients P1-3 of family 1, P4-5 of family 2, P8 of family 3, and P9, P10, and P11, all of whom suffered from asthenoteratozoospermia (*Figure 1A*). Initially, a homozygous frameshift mutation c.944_945 del (p. Phe315*) in family 1, and two homozygous missense mutations, c. A164G (p. Gln55Arg) in family 2 and c. A2560G (p. Ile854Val) in family 3, were identified in *DNAH12* as potential causes of infertility. Subsequently, four other mutations in *DNAH12* were discovered in three sporadic Chinese patients with asthenoteratozoospermia: a homozygous stop-gain mutation c.2680 C>T (p. Gln2077*) in subject P9, a homozygous missense mutation c.2813T>C (p. Val938Ala) in P10, and compound heterozygous mutations c.2680C>T (p. Arg894Cys) and c.5964–3A>G in P11 (*Figure 1A and B*).

All seven variants were rare in public human genome databases, including the 1000 Genome Project and gnomAD (v3). Additionally, these variants were predicted to be deleterious by in-silico software tools such as SIFT, Polyphen2, fathmm-MKL, and CADD2 (*Table 1*). Sanger Sequencing further validated these variants and revealed the recessive pathogenic pattern of these mutations (*Figure 1A*). Collectively, these results suggested that the identified bi-allelic *DNAH12* variants could potentially be the cause of asthenoteratozoospermia in our patients.

### DNAH12 protein is conserved in mammals and highly expressed in the testes

*DNAH12* encodes a functionally uncharacterized protein and is predicted to be abundantly expressed in testes of humans and mice based on our bioinformatics tool FertilityOnline (*Gao et al., 2022b*). Experimentally, our RT-PCR, qPCR, and immunoblotting detection of DNAH12 expression in different tissues indicated that both DNAH12 mRNA and protein showed high expression in testes and relatively lower expression in lungs and tracheas of male mice and oviducts of female mice (*Figure 1— figure supplement 2A–C*). It's worth noting that in epididymides, DNAH12 protein, but not mRNA, was detected, indicating the DNAH12 protein detected in epididymides is from testes, most probably from sperm. Furthermore, DNAH12 protein was initially detected in testes around 21 d postpartum (dpp) and thereafter maintained a high level (*Figure 1—figure supplement 2D*). The expression pattern of DNAH12 in testes suggests that it is possibly involved in spermiogenesis, and its abnormal expression could lead to asthenozoospermia or asthenoteratozoospermia, further indicating that the identified *DNAH12* mutations are candidate pathogenic mutations for patients.

Further phylogenetic analysis revealed the conservative nature of DNAH12 from Chlamydomonas to humans (*Figure 1—figure supplement 3*). All the amino acids affected by mutations were conserved in mammals and were located in functional regions, including the ATPase domains or the Stem domain of DNAH12 (*Figure 1C*), indicating that these mutations are highly likely to hinder the structure, function, or stability of the DNAH12 protein. Altogether, our results suggested that the identified mutations in *DNAH12* are probable pathogenic factors contributing to asthenoteratozoospermia in our patients.

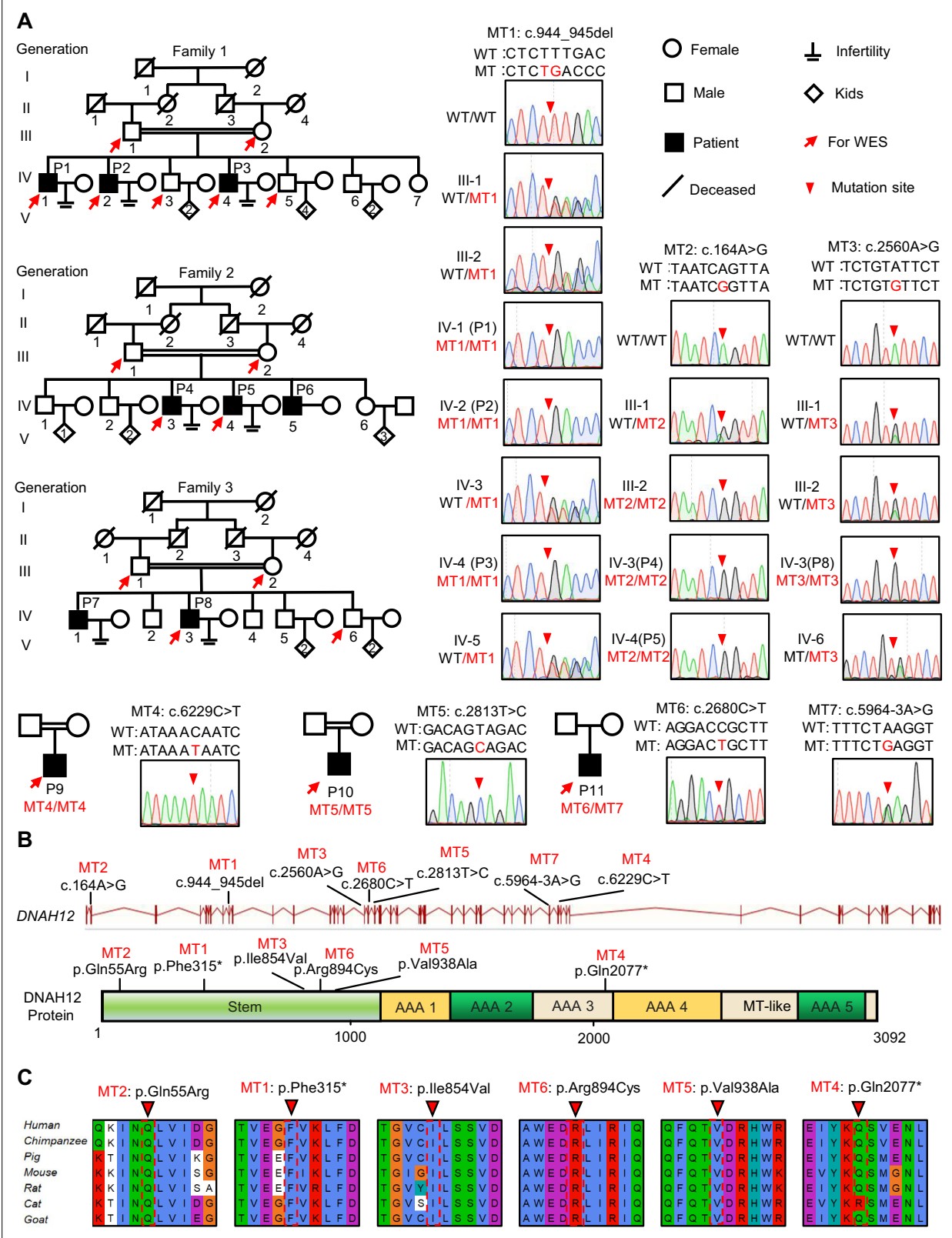

**Figure 1.** Identification of bi-allelic *DNAH12* variants in infertile men with asthenoteratozoospermia. (**A**) Pedigrees of family 1 with three infertile males, P1 (IV:1), P2 (IV:2), and P3 (IV:4), family 2 with three infertile males, P4 (IV:3), P5 (IV:4) and P6 (IV:5), family 3 with two infertile males, P7 (IV:1) and P8 (IV:3), and three sporadic infertile cases P9, P10 and P11. Red arrows point to the individuals for whom whole-exome sequencing was performed. Candidate mutations were validated by Sanger sequencing and the chromatograms were shown on the right. Red arrowheads point to mutant sites. Double

*Figure 1 continued on next page*

*Figure 1 continued*

horizontal lines represent consanguineous marriages. WT, wild-type allele; MT, the mutant allele. (**B**) The positions of the *DNAH12* variants at the transcript (ENST00000351747.2) and protein levels (Q6ZR08-1). MT-like, Microtubule-binding-like domain. (**C**) Conservation analyses of affected amino acids across different species, the arrowheads indicate the mutation sites.

The online version of this article includes the following source data and figure supplement(s) for figure 1:

**Figure supplement 1.** Flowchart for analyses of the whole-exome sequencing data.

**Figure supplement 2.** Expression pattern of *Dnah12* in different mouse tissues and developmental stages of the testis.

**Figure supplement 2—source data 1.** Labelled files for western blot in *Figure 1—figure supplement 2*.

**Figure supplement 2—source data 2.** Original files for western blot in *Figure 1—figure supplement 2*.

**Figure supplement 2—source data 3.** Labelled file for gel in *Figure 1—figure supplement 2*.

**Figure supplement 2—source data 4.** Original file for gel in *Figure 1—figure supplement 2*.

**Figure supplement 3.** Phylogenetic analysis of the DNAH12 homologous proteins in different species.

## Impaired sperm motility and morphology in patients with bi-allelic *DNAH12* variants

To assess the pathological effects of the identified bi-allelic *DNAH12* variants, semen analyses were conducted following WHO guidelines (*WHO, 2021*). All affected individuals exhibited significantly reduced sperm motility and progressive motility (*Table 2*). Detailed sperm morphology analyses were performed on patients P3 from family 1, P5 from family 2, P8 from family 3, and three Chinese sporadic patients (P9, P10, and P11). Spermatozoa from fertile controls displayed normal heads and tails, while spermatozoa from the patients exhibited a range of abnormal morphologies, including short, coiled, absent, bent, or irregular caliber flagella, accompanied by aberrant heads (*Figure 2A*, *Figure 2— figure supplement 1*). Scanning electron microscopy (SEM) analyses were further carried out on a fertile control and P10, confirming the presence of abnormal morphologies of the sperm head and flagellum in the patients (*Figure 2B*). Thus, patients with the bi-allelic *DNAH12* variants demonstrated significantly decreased sperm motility and increased abnormal sperm morphology.

## The bi-allelic *DNAH12* variants result in absence of DNAH12 protein in patients' sperm

To investigate the effects of the identified bi-allelic *DNAH12* variants on the DNAH12 protein, we obtained sperm samples from three affected Chinese patients, and the expression of DNAH12 was detected using a custom-made DNAH12 rabbit antibody that recognized the N-terminal 1–200 amino

**Table 1.** Detailed information of bi-allelic *DNAH12* variants identified in infertile men.

The accession number of human *DNAH12* is ENST00000351747.2. (b) Abbreviations are listed as follows: SNV, single nucleotide variants; NA, not assessed; T, tolerated; D, deleterious. SIFT, Sorting Intolerant From Tolerant; PolyPhen-2, Polymorphism Phenotype v2; FATHMM-MKL, Functional Analysis Through Hidden Markov Models-Multiple Kernel Learning; CADD, Combined Annotation-Dependent Depletion; and a mutation is predicted deleterious if the CADD Phred score >20 in CADD. gnomAD, Genome Aggregation Database.

| | Genomic position on chr3 | cDNA Change a | Mutation type | Genetic pattern | SIFT | Polyphen2 | fathmm-MKL | CADD | 1000 Genomes | gnomAD |
|---|---|---|---|---|---|---|---|---|---|---|
| Family 1 | 57,489,883 | MT1: c.944_945del | frameshift substitution | AR | NA | NA | NA | NA | 0 | 0.00002844 |
| Family 2 | 57,528,434 | MT2: c.164A>G | nonsynonymous SNV b | Male-limited AR | T | D | D | D | 0.000599042 | 0.0004 |
| Family 3 | 57,447,323 | MT3: c.2560A>G | nonsynonymous SNV | AR | T | D | D | D | 0.000599042 | 0.0002 |
| P9 | 57,391,670 | MT4: c.6229C>T | stop-gain | AR | NA | NA | D | NA | 0 | 0.00002714 |
| P10 | 57,445,368 | MT5: c.2813T>C | nonsynonymous SNV | AR | D | D | D | D | 0.000199681 | 0.00003383 |
| | 57,394,265 | MT6: c. 5964−3A>G | splicing | | NA | NA | NA | NA | 0.000199681 | 0.0002 |
| P11 | 57,445,501 | MT7: c.2680C>T | nonsynonymous SNV | AR | D | D | D | D | 0 | 0.000007178 |

**Table 2.** Clinical characteristics of the infertile patients with bi-allelic *DNAH12* mutations.

The reference limits are shown according to WHO sixth edition standards. (b) Reference values are shown by observation of sperm morphology in three fertile individuals. (c) 'Age' represents age in 2024. ND, not determined.

| | Reference values [a] | Fertile controls [b] | Family1 | | | Family2 | | Family3 | | | |
|---|---|---|---|---|---|---|---|---|---|---|---|
| | | | IV-1(P1) | IV-2(P2) | IV-4(P3) | IV-5(P4) | IV-7(P5) | IV-4(P8) | P9 | P10 | P11 |
| cDNA mutation | – | – | MT1/MT1 | MT1/MT1 | MT1/MT1 | MT2/MT2 | MT2/MT2 | MT3/MT3 | MT4/MT4 | MT5/MT5 | MT6/MT7 |
| Age (y) c | – | – | 58 | 49 | 42 | 40 | 38 | 32 | 33 | 40 | 31 |
| **Semen parameters** | | | | | | | | | | | |
| Semen volume (ml) | >1.4 | – | ND | 1.5±0.5 | 2.8±0.3 | 4.0 | 3.0 | 3.0 | 3.2 | 3.1 | 2.4 |
| Sperm concentration ($10^6$ /ml) | >15 | – | ND | 58±28.5 | 50.0±15.0 | 6.0 | 8.0 | 2.8 | 24.7 | 83.1 | 57.1 |
| Semen pH | Alkaline | – | ND | Alkaline | Alkaline | Alkaline | Alkaline | Alkaline | Alkaline | Alkaline | Alkaline |
| Motile sperm (%) | >40 | – | ND | 29.3±9.0 | 7.5±7.5 | 14.0 | 20.0 | 20 | 15.1 | 17.4 | 7.4 |
| Progressively motile sperm (%) | >30 | – | ND | 13.3±10.3 | 16.7±17 | 5.0 | 8.0 | 5 | 13.4 | 16.8 | 24 |
| **Sperm morphology** | | | | | | | | | | | |
| Normal (%) | – | 76.5±2.0 | ND | ND | 19.9 | ND | 21.7 | 17.0 | 14.0 | 17.4 | 7.4 |
| Abnormal head (%) | – | 7.0±1.9 | ND | ND | 16.4 | ND | 5.4 | 6.0 | 3.5 | 16.4 | 20.0 |
| Abnormal tail (%) | – | 12.4±3.4 | ND | ND | 43.8 | ND | 41.4 | 44.0 | 42.4 | 37.4 | 16.3 |
| Abnormal head &tail (%) | – | 4.1±1.3 | ND | ND | 19.9 | ND | 31.5 | 33.0 | 40.2 | 28.8 | 56.3 |
| **Sperm flagella morphology** | | | | | | | | | | | |
| Normal (%) | – | 83.1±2.3 | ND | ND | 26.4 | ND | 26.6 | 22.7 | 17.9 | 31.7 | 27.4 |
| Absent (%) | – | 2.7±1.1 | ND | ND | 16.4 | ND | 7.9 | 15.8 | 9.6 | 16.6 | 11.6 |
| Short (%) | – | 3.6±0.6 | ND | ND | 16.4 | ND | 11.3 | 14.9 | 20.1 | 11.7 | 8.4 |
| Coiled (%) | – | 2.8±0.7 | ND | ND | 17.3 | ND | 24.6 | 12.9 | 11.8 | 14.6 | 7.0 |
| Bent (%) | – | 6.3±0.6 | ND | ND | 12.7 | ND | 15.8 | 10.0 | 13.5 | 7.8 | 33.5 |
| Irregular caliber (%) | – | 1.5±0.2 | ND | ND | 10.9 | ND | 13.8 | 23.8 | 27.1 | 17.6 | 12.1 |

acids of the DNAH12 protein. The validated antibody could theoretically detect the truncated protein if it were present in P9 with a homozygous stop-gain mutation M4 (p. Gln2077*). The predicted DNAH12 protein size was detected in the sperm lysate from a fertile control but not in P9 with the homozygous stop-gain mutation M4, P10 with a homozygous missense mutation M5, and P11 with compound heterozygous mutations M6 and M7, indicating these mutations possibly lead to the disruption of the DNAH12 protein (*Figure 2C*).

To further explore the effects of the identified *DNAH12* variants on the DNAH12 protein, we performed immunofluorescence assays on sperm from patients. The results showed that DNAH12 signals were localized along the flagella of the fertile control, however, no DNAH12 signals were observed in sperm flagella of patients P3 (family 1), P5 (family 2), P8 (family 3), P9, P10, and P11 (*Figure 2D*). Collectively, these findings indicated that all the detected *DNAH12* variants cause the absence of DNAH12 protein in the sperm of our patients, reinforcing that these *DNAH12* mutations are pathogenic mutations of asthenoteratozoospermia in the patients.

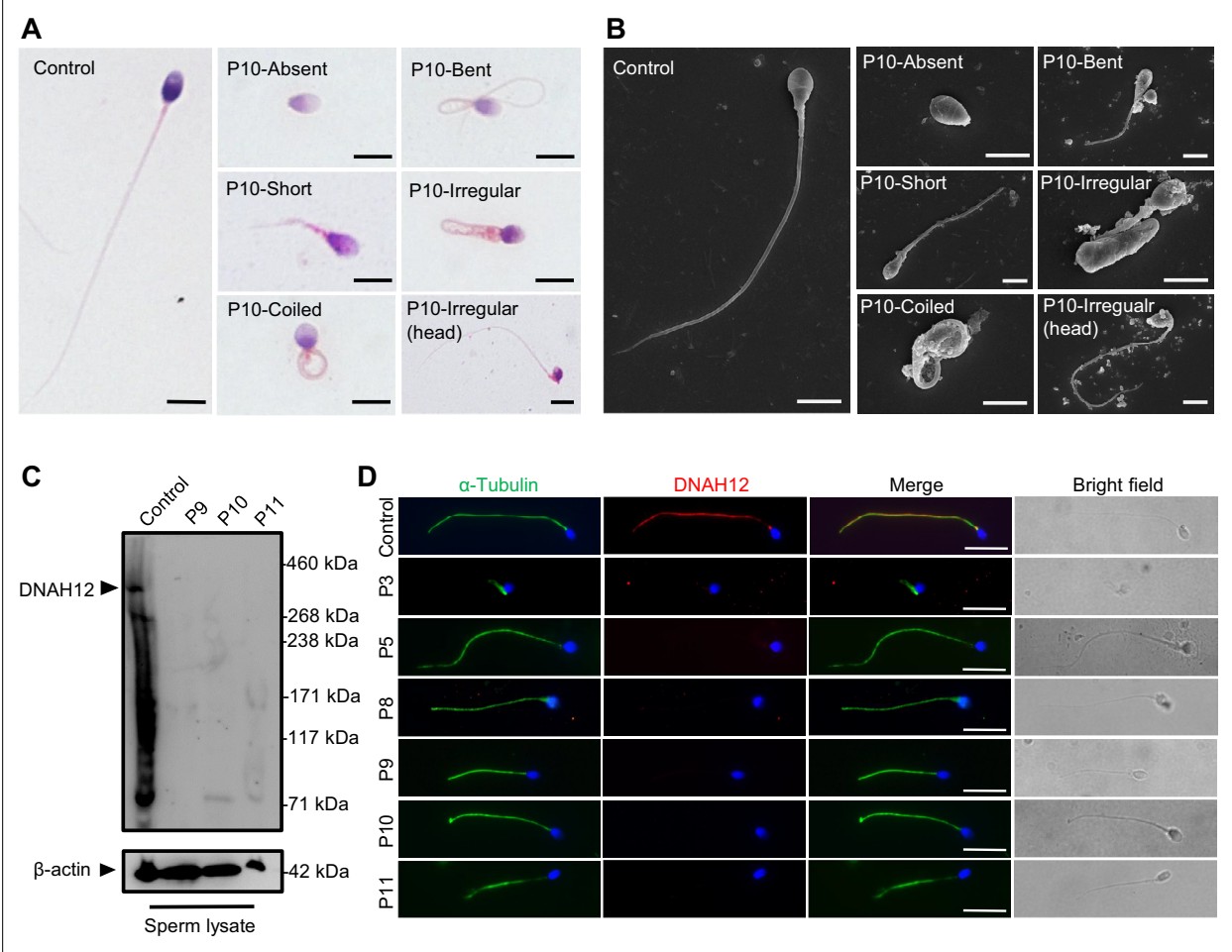

**Figure 2.** Abnormal sperm morphology and undetectable DNAH12 expression in men with bi-allelic *DNAH12* variants. (**A**) Representative micrographs show normal sperm morphology from a fertile control and abnormal spermatozoa from P10 with absent, short, coiled, bent, irregular flagella, or abnormal head under light microscopy. Scale bars, 5 μm. (**B**) Representative scanning electron microscopy (SEM) micrographs show details about the sperm morphology of a fertile control and P10. The normal sperm morphology can be observed in the fertile control (left) and abnormal types like absent, short, coiled, bent, irregular tails, and head abnormality in P10 (right). Scale bars, 5 μm. (**C**) Immunoblotting assay revealed that DNAH12 was absent in the spermatozoa from P9, P10, and P11 harboring *DNAH12* variants. β-actin was used as a loading control. (**D**) Sperm cells were co-stained with α-Tubulin (green) and DNAH12 (red) antibodies while no DNAH12 signals were observed in sperm flagella of patients. DNA was counterstained with Hoechst 33342. Scale bars, 10 μm.

The online version of this article includes the following source data and figure supplement(s) for figure 2:

**Source data 1.** Labelled file for western blot in *Figure 2A*.

**Source data 2.** Original file for western blot in *Figure 2A*.

**Figure supplement 1.** Sperm morphology of fertile controls and patients.

## Patients with bi-allelic *DNAH12* variants exhibit flagellar axoneme defects

Since patients with the bi-allelic *DNAH12* variants exhibited compromised sperm motility and abnormal sperm morphology, we then investigated whether sperm flagellar ultrastructure was also affected in these patients. TEM examination was performed on sperm flagella from both patients and normal fertile controls. The flagella of fertile controls displayed the typical '9+2' microtubule structure with well-organized DMTs and CP. In contrast, various defects were found in our patients, including loss of CP (54.5%), disorganization of microtubules (26%), and absence of DMTs (16.5%) in the examined cross-sections (*Figure 3A, C*, *Figure 3—figure supplement 1A*). To examine the IDAs impairment in axonemes, the sections with intact ODAs were analyzed. Compared with axonemes in controls that almost all the sections presented intact IDAs, approximately 70% of sections examined showed

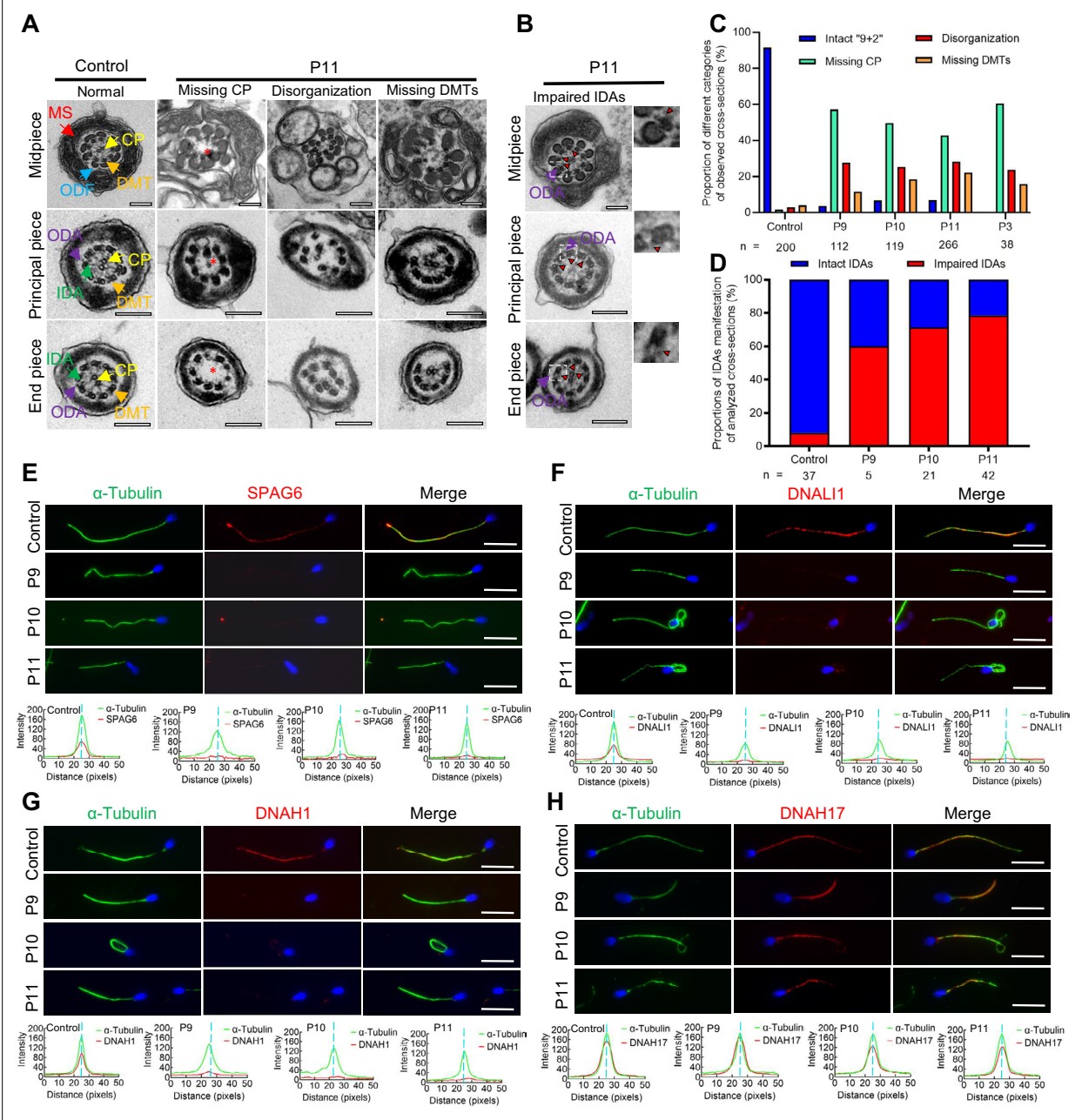

**Figure 3.** Flagellar axoneme defects were detected in the patients with *DNAH12* variants. (**A–B**) Representative transmission electron microscopy (TEM) micrographs showing cross-sections of the midpiece, principal piece, and end piece of sperm flagella from a fertile control and P11. The axoneme structure in control presents a '9+2' microtubules arrangement, including mitochondrial sheath (MS, indicated with red arrows), central pair of microtubules (CP, indicated with yellow arrows), outer dense fibers (ODFs, indicated with cerulean arrows), peripheral doublet microtubules (DMTs, indicated with orange arrows), inner dynein arms (IDAs, indicated with green arrows), and outer dynein arms (ODAs, indicated with purple arrows), while in P11, axonemal defects like missing CP, disorganized axonemal structures or missing DMTs were observed, the red asterisks mark CP loss (**A**). Impaired IDAs in the sperm axoneme of patient P11 (**B**). The red triangle marks impaired IDAs while adjacent ODAs are identifiable. Scale bars, 200 nm. (**C–D**) The proportion of different categories of observed cross-sections in the control and patients. Cross-sections were classified into four categories: Intact '9+2', missing CP, disorganization, and missing DMTs (**C**), and the proportion of intact IDAs or impaired IDAs in the cross-sections of flagellar axoneme of which ODAs were identifiable (**D**). n, the total number of cross-sections for quantification. (**E–G**) Representative images of spermatozoa from fertile controls and patients carrying bi-allelic *DNAH12* variants co-stained α-Tubulin with SPAG6 (**E**), DNAH1 (**F**), DNALI1 (**G**), or DNAH17 (**H**), and Hoechst 33342 for DNA (blue). The fluorescent signal intensity profiles were shown on the bottom. Scale bars, 10 μm.

The online version of this article includes the following figure supplement(s) for figure 3:

**Figure supplement 1.** Transmission electron microscopy (TEM) images of men harboring bi-allelic *DNAH12* variants and immuno-fluorescence assay of ODF2 in sperm of control, P9, P10 and P11.

impaired IDA structures in the analyzed three patients (*Figure 3B, D*, *Figure 3—figure supplement 1A*).

We then performed immunofluorescence assays on sperm smears to assess the expression and localization of specific proteins of the axonemes. These included SPAG6 (a CP component), DNAH1 and DNALI1 (IDA components), DNAH17 (an ODA component), and ODF2 (an outer dense fiber, ODF component). The sperm flagella of patients showed remarkably reduced SPAG6 signals when compared with those from controls (*Figure 3E*). Similarly, DNAH1 and DNALI1 signals were both obviously decreased in the patients' sperm flagella, indicating impaired IDAs in the patients (*Figure 3F and G*). Besides, DNAH17 and ODF2 signals in patients' flagella were comparable to those in a normal control (*Figure 3H*, *Figure 3—figure supplement 1B*). In summary, these findings indicated that the bi-allelic *DNAH12* variants cause pronounced flagellar axoneme defects, including impaired IDAs and loss of CP, but do not affect ODAs and ODFs.

## Deletion of *Dnah12* impairs spermiogenesis and causes male infertility in mice

Considering the bi-allelic *DNAH12* variants resulting in absence of DNAH12 protein in patients' sperm, to functionally confirm the deletion of *DNAH12*, we generated *Dnah12* knockout mice using CRISPR-Cas9 approach (*Figure 4—figure supplement 1A*). Fortunately, we obtained a mouse model with a 4-bp deletion (c.386_389del) in exon 5 of *Dnah12* (*Figure 4—figure supplement 1A*). In contrast to the observations in control mice, immunoblotting of testis lysate using two antibodies targeting the N-terminal of DNAH12 failed to detect DNAH12 proteins, and immunofluorescence staining of sperm flagella did not show DNAH12 signals in *Dnah12$^{-/-}$* mice (*Figure 4—figure supplement 1B* and C). *Dnah12$^{-/-}$* males exhibited normal development but were infertile (*Figure 4A*), while *Dnah12$^{-/-}$* female mice showed normal development and fertility, and no obvious defective oviductal and tracheal cilia (*Figure 4—figure supplement 2A–C*). The body weight of *Dnah12$^{-/-}$* and *Dnah12$^{+/+}$* males was comparable, while smaller testes were observed in *Dnah12$^{-/-}$* males (*Figure 4B*). Besides, the average number of sperm per epididymis of *Dnah12$^{-/-}$* mice dramatically decreased to approximately 2% of that of *Dnah12$^{+/+}$* mice (*Figure 4C*). The epididymal caudal spermatozoa of *Dnah12$^{-/-}$* male mice were 100% immotile, compared to those of *Dnah12$^{+/+}$* mice (*Figure 4D*, *Videos 1 and 2*). H&E staining of testis and epididymis sections also showed a striking reduction in the number of sperm in *Dnah12$^{-/-}$* mice (*Figure 4E*). Importantly, morphological analysis of sperm from the caudal epididymis of *Dnah12$^{-/-}$* mice showed similar types and frequencies of flagellum and head abnormalities to those observed in our patients (*Figure 4F and G*).

To understand how *DNAH12* depletion causes abnormal sperm morphology, we conducted immunofluorescence assays on testicular cells from *Dnah12$^{+/+}$* and *Dnah12$^{-/-}$* mice, in *Dnah12$^{-/-}$* mice, while DNAH12 signals are missing in round, elongating, and elongated spermatids; the abnormal manchette structure and flagellum were observed (*Figure 4H*). We then performed immunofluorescence assays on testicular sections using α-Tubulin and lectin peanut agglutinin (PNA, a marker of the acrosome) to examine the process of spermiogenesis. The manchette structure in *Dnah12$^{-/-}$* was roughly normal until steps 9–10 (Stage IX-X) but was aberrantly stretched from steps 11–12 (Stage XI-XII) when compared with those of *Dnah12$^{+/+}$* mice (*Figure 4—figure supplement 3*). To meticulously characterize these defects, we stained α-Tubulin and PNA on squashed seminiferous tubules and found obvious manchette defects and abnormal sperm heads in *Dnah12$^{-/-}$* mice (*Figure 4I*). TEM analysis showed severely abnormal head morphology in elongated spermatids of *Dnah12$^{-/-}$* males when compared with those of *Dnah12$^{+/+}$* males (*Figure 4J*). Additionally, TUNEL assays of testis sections showed dramatically increased proportions of seminiferous tubules containing TUNEL-positive cells in *Dnah12$^{-/-}$* mice, compared with those in *Dnah12$^{+/+}$* mice, suggesting the aberrant spermatids may undergo apoptosis (*Figure 4K–M*). Altogether, these results indicated that deletion of *Dnah12* impairs spermiogenesis, results in decreased sperm counts, and increased sperm morphological abnormalities in mice.

## Deletion of *Dnah12* causes axoneme defects in sperm flagella but not cilia in mice

TEM analysis was conducted on sperm from cauda epididymis of *Dnah12$^{+/+}$*, *Dnah12$^{+/-}$* and *Dnah12$^{-/-}$* mice. Flagellar axonemes of *Dnah12$^{+/+}$* and *Dnah12$^{+/-}$* mice displayed a typical '9+2' microtubular arrangement with clearly visible CP, IDAs, and ODAs. In contrast, sperm from *Dnah12$^{-/-}$* mice

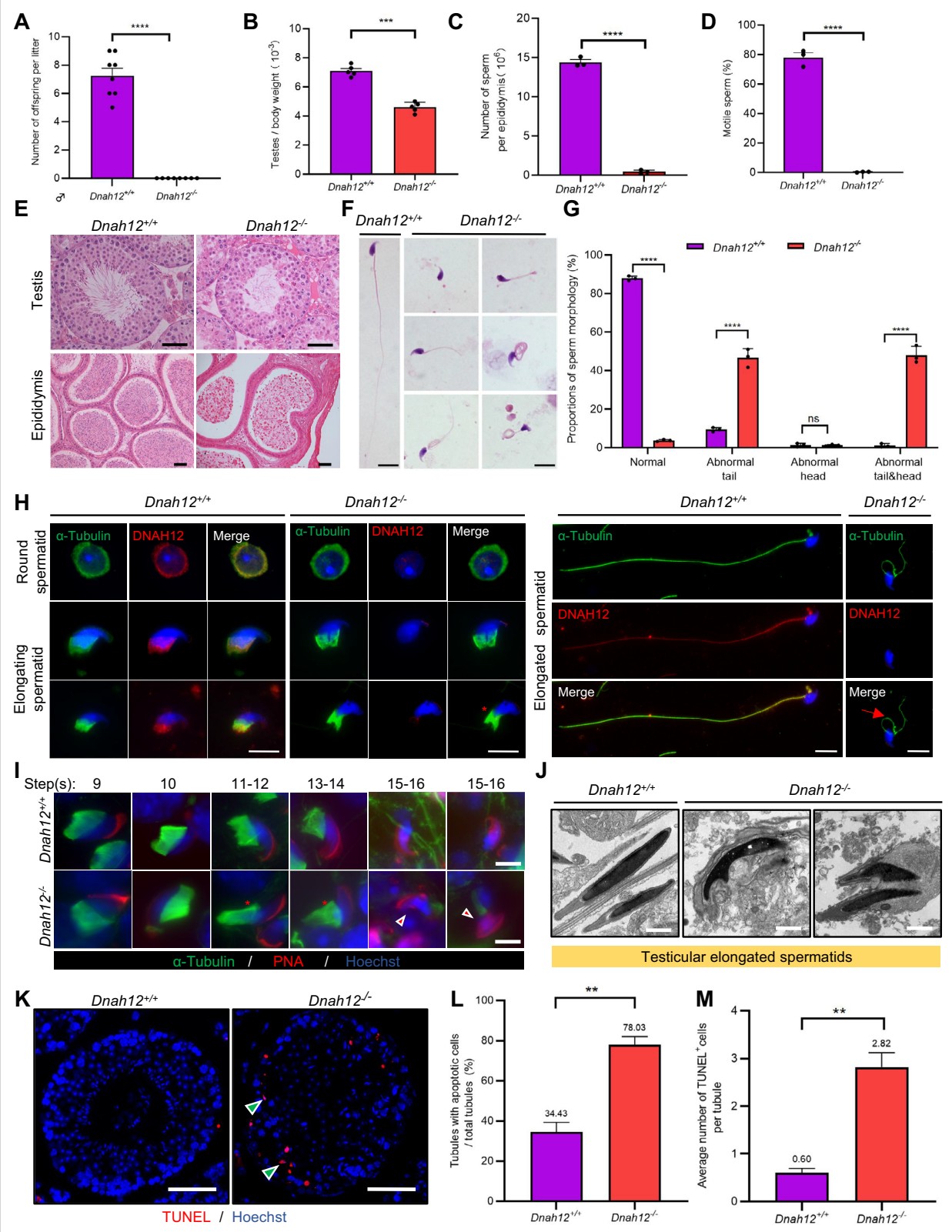

**Figure 4.** Disruption of *Dnah12* results in impaired spermatogenesis and abnormal spermatozoa morphology. (**A**) Fertility test of adult *Dnah12+/+* and *Dnah12-/-* male mice (n=8 independent experiments). Data are presented as mean ± SEM; ****p<0.0001. (**B**) The testes to body weight ratios of *Dnah12+/+ and Dnah12-/-* mice (n=5 independent experiments). Data are presented as mean ± SEM; ***p<0.001. (**C**) The number of sperm per epididymis of *Dnah12+/+ and Dnah12-/-* male mice (n=3 independent experiments). Data are presented as mean ± SEM; ****p<0.0001. (**D**) Percentages

*Figure 4 continued on next page*

*Figure 4 continued*

of motile spermatozoa in *Dnah12*^(+/+) *and Dnah12*^(-/-) male mice (n=3 independent experiments). Data are presented as mean ± SEM; ****p<0.0001. (**E**) Histological sections of testis and epididymis from *Dnah12*^(+/+) and *Dnah12*^(-/-) mice after H&E staining. Scale bars, 50 μm. (**F**) Morphology of the sperm from caudal epididymis of *Dnah12*^(+/+) and *Dnah12*^(-/-) mice. Scale bars,10 μm. (**G**) Quantitative analysis of sperm morphology of *Dnah12*^(+/+) *and Dnah12*^(-/-) male mice. The experiments were repeated three times with at least 200 spermatozoa counted every time. The above data were obtained from three adult mice for each genotype (n=3 independent experiments). Data are presented as mean ± SEM; ns indicates no significant difference; ****p<0.0001. (**H**) Representative images of testicular cells from *Dnah12*^(+/+) and *Dnah12*^(-/-) mice co-stained DNAH12 and α-Tubulin antibodies. Scale bars, 10 μm. The red asterisk indicates abnormal manchette structure and the red arrow indicates abnormal sperm flagellum at the elongated spermatid stage of *Dnah12*^(-/-) mice. (**I**) Representative immunofluorescence images of seminiferous tubule squash from 8-wk-old *Dnah12*^(+/+) and *Dnah12*-/- mice, co-stained by peanut agglutinin (PNA) (red) and α-Tubulin (green) antibody. DNA was stained with Hoechst 33342. Scale bars, 10 μm. The asterisks indicate the manchette defects and the red triangles mark the abnormal sperm heads. Scale bars,10 μm. (**J**) Representative transmission electron microscopy (TEM) micrographs of testicular elongated spermatids from *Dnah12*^(+/+) and *Dnah12*^(-/-) mice. Scale bars, 1 μm. (**K**) TUNEL assay on the testicular sections from *Dnah12*^(+/+) and *Dnah12*^(-/-) mice. The green triangles mark the apoptotic elongating or elongated spermatids. Scale bars, 50 μm. (**L**) The proportion of tubules with apoptotic cells in *Dnah12*^(+/+) and *Dnah12*^(-/-) mice. (**M**) Average numbers of TUNEL-positive cells per tubule in *Dnah12*^(+/+) and *Dnah12*^(-/-) mice (n=3 independent experiments). Data are obtained from three mice with at least 50 tubules scored in each repeated experiment. Data are presented as mean ± SEM; **p<0.01.

The online version of this article includes the following source data and figure supplement(s) for figure 4:

**Figure supplement 1.** Generation of *Dnah12*^(-/-) mice model and validation of rabbit and rat host DNAH12 antibodies.

**Figure supplement 1—source data 1.** Labelled files for western blot in *Figure 4—figure supplement 1*.

**Figure supplement 1—source data 2.** Original files for western blot in *Figure 4—figure supplement 1*.

**Figure supplement 2.** Normal development and fertility in *Dnah12*^(-/-) female mice.

**Figure supplement 3.** Deficiency in manchette organization and abnormal sperm head shaping during spermiogenesis in *Dnah12*^(-/-) mice.

exhibited severe abnormalities in all the analyzed sections, characterized by CP loss and disorganized axoneme structure (**Figure 5A**). Immunoblotting was performed to assess the expression levels of DNAH1, DNALI1, and SPAG6 in sperm lysates, *Dnah12*^(-/-) sperm showed a significant reduction in the abundance of DNAH1 and DNALI1, while the amount of DNAI2 (a component of ODAs) remained comparable to *Dnah12*^(+/+) and *Dnah12*^(+/-) sperm (**Figure 5B**). Additionally, immunofluorescence assays revealed the near absence of signals for SPAG6, SPEF2 (a component of CP complex), DNAH1, and DNALI1 in *Dnah12*^(-/-) sperm flagella, whereas DNAH17 and DNAI2 (components of ODAs) remained unaffected (**Figure 5C–H**). These results indicate that DNAH12 deficiency specifically impairs CP and IDAs, but not ODAs, in sperm flagella. Furthermore, immunoblotting of SPAG6 and DNALI1 in testis lysate exhibited a sharp decrease in *Dnah12*^(-/-) mice (**Figure 5—figure supplement 1A**). Additionally, TEM examination of testicular sperm axonemes showed defects like loss of CP and disarranged axonemes in *Dnah12*^(-/-) mice (**Figure 5—figure supplement 1B**). These results indicate that impaired IDAs and loss of CP possibly occur during testicular sperm flagellar development.

Another *Dnah12* mutant mouse model (*Dnah12*^(mut/mut)) targeting exon18 was also generated (**Figure 5—figure supplement 2A**), the *Dnah12*^(mut/mut) mice may produce a truncated DNAH12 protein (p.Cys793Glu, fs*10) which could distinguish from *Dnah12*^(-/-) mice. The mutation effect was checked but no truncated DNAH12 protein was detected, indicating that DNAH12 expression was disrupted in *Dnah12*^(mut/mut) mice (**Figure 5—figure supplement 2B**). Moreover, *Dnah12*^(mut/mut) mice developed normally, while decreased testicular size was observed (**Figure 5—figure supplement 2C and D**).

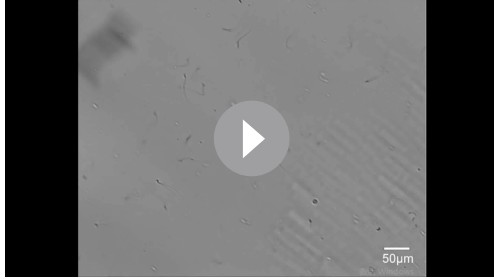

**Video 1.** Sperm motility of caudal sperm from *Dnah12*^(+/+) mice.

https://elifesciences.org/articles/100350/figures#video1

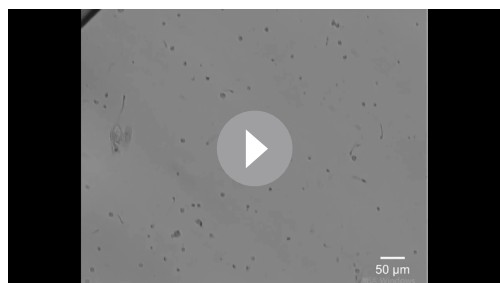

**Video 2.** Sperm motility of caudal sperm from *Dnah12*^(-/-) mice.

https://elifesciences.org/articles/100350/figures#video2

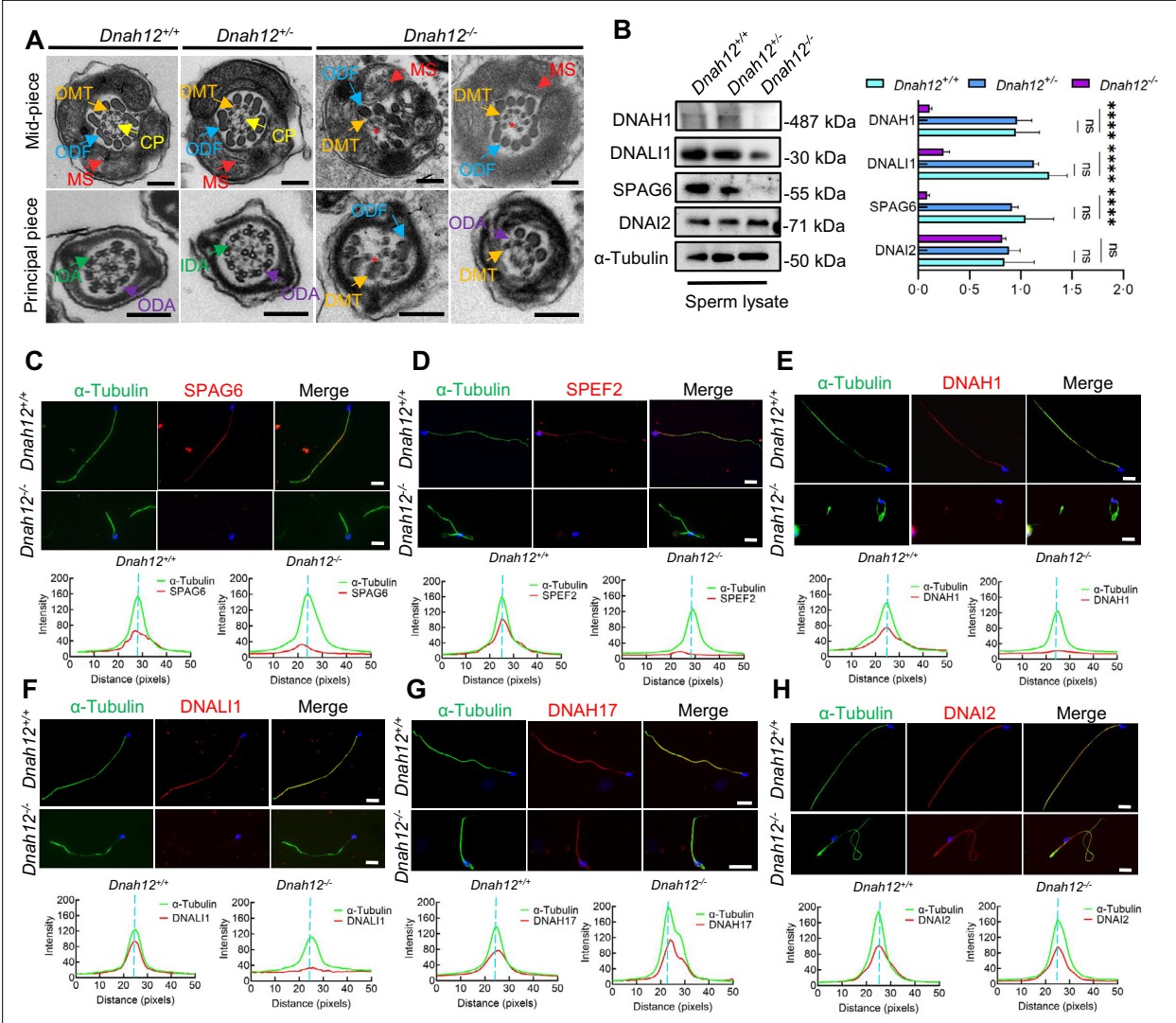

**Figure 5.** Deletion of *Dnah12* causes axoneme defects in mice. (**A**) Representative transmission electron microscopy (TEM) micrographs showing cross-sections of sperm flagella from *Dnah12⁺/⁺*, *Dnah12⁺/⁻*, and *Dnah12⁻/⁻* mice. In *Dnah12⁺/⁺*, *Dnah12⁺/⁻*, the normal axoneme '9+2' microtubule arrangement mainly consisted of the mass spectrometry (MS), (indicated with red arrows), central pair (CP) (yellow arrows), outer dense fibers (ODFs) (cerulean arrows), doublet microtubules (DMTs) (orange arrows), inner dynein arms (IDAs) (green arrows), and outer dynein arms (ODAs) (purple arrows). However, the CP and DMTs structures were missing or disarranged in the *Dnah12⁻/⁻* group. Red asterisks mark the loss of CP. Scale bars, 200 nm. (**B**) Immunoblotting of sperm lysate from *Dnah12⁺/⁺*, *DNAH12⁺/⁻* and *Dnah12⁻/⁻* mice using DNAH1, DNALI1 or SPAG6 antibodies. DNAI2 and α-Tubulin were used as the loading controls. The relative band grayscales of proteins to α-Tubulin in sperm lysate from *Dnah12⁺/⁺*, *DNAH12⁺/⁻*, and *Dnah12⁻/⁻* mice were shown on the right (n=4 independent experiments). Data are presented as mean ± SEM; ns indicates no significant difference; ****p<0.0001. (**C–H**) Representative images of caudal epididymal sperm from *Dnah12⁺/⁻* and *Dnah12⁻/⁻* mice co-stained α-Tubulin and SPAG6 (**C**), SPEF2 (a component of CP complex) (**D**), DNAH1 (**E**), DNALI1 (**F**), DNAH17 (**G**) or DNAI2 (**H**) antibodies. The fluorescent signal intensity profiles were shown on the bottom. Scale bars, 10 μm.

The online version of this article includes the following source data and figure supplement(s) for figure 5:

**Source data 1.** Labelled files for western blot in *Figure 5B*.

**Source data 2.** Labelled files for western blot in *Figure 5B*.

**Figure supplement 1.** The central pair (CP) and doublet microtubules (DMTs) structures were impaired in the *Dnah12⁻/⁻* testicular sperm axoneme.

**Figure supplement 1—source data 1.** Labelled files for western blot in *Figure 5—figure supplement 1*.

**Figure supplement 1—source data 2.** Original files for western blot in *Figure 5—figure supplement 1*.

**Figure supplement 2.** Generation of *Dnah12^{mut/mut}* mice.

**Figure supplement 2—source data 1.** Labelled files for western blot in *Figure 5—figure supplement 2*.

*Figure 5 continued on next page*

*Figure 5 continued*

**Figure supplement 2—source data 2.** Original files for western blot in *Figure 5—figure supplement 2*.

**Figure supplement 3.** Normal development and no obvious primary ciliary dyskinesia (PCD) symptoms were observed in *Dnah12^mut/mut* and *Dnah12^-/-* males.

Consistent with *Dnah12^-/-* mice and patients carrying bi-allelic *DNAH12* mutations, *Dnah12^mut/mut* presented obviously decreased sperm count, increased sperm morphological abnormalities, and sperm axoneme defects similar to *Dnah12^-/-* mice (*Figure 5—figure supplement 2E–H*). Therefore, these results reconfirmed that DNAH12 is essential for sperm flagellar development.

Given the detection of DNAH12 protein in the trachea and lung (*Figure 1—figure supplement 2A–C*), we also examined whether the deficiency of DNAH12 would cause PCD symptoms in males. Intriguingly, both *Dnah12^mut/mut* and *Dnah12^-/-* males developed normally and showed no tracheal motility or structure defects, despite the expression of DNAH12 being detected in tracheal cilia cells (*Figure 5—figure supplement 3A–E*; *Videos 3 and 4*). Besides, the ultra-structures of cilia axonemes were comparable in wild-type and *Dnah12^-/-* groups (*Figure 5—figure supplement 3F*). These findings collectively suggested that DNAH12 deficiency is crucial for sperm axonemal integrity in humans and mice without manifesting PCD symptoms.

## DNAH12 interacts with IDA components DNAH1 and DNALI1

Through phylogenetic and domain analyses of DNAH family members, we found that DNAH12 is a distinctive IDA lacking the microtubule-binding domain (MTBD), which is crucial for ATP-sensitive microtubule binding and motor direction control (*Figure 6A*, *Figure 6—figure supplement 1*). Therefore, DNAH12 likely collaborates with other members to carry out its functions. To delve into the molecular mechanism of DNAH12, we conducted co-immunoprecipitation (Co-IP) followed by mass spectrometry analysis using a DNAH12 rabbit antibody in testicular lysate from adult *Dnah12^+/+* mice (*Figure 6B*). Gene ontology analysis showed that the potential interactors were mainly enriched in motor protein, actin binding, myosin and RNA binding proteins (*Figure 6C*). The motor proteins were predominantly categorized into three groups: dynein, kinesin, and myosin protein members, interestingly, among these, dynein proteins such as DNAH1, DNALI1, DNAH1, DNAH3, and DNAH2 were annotated to be involved in IDA assembly, which is closely related to the defects found in *DNAH12* mutants (*Figure 6D*). This piqued our interest to explore these proteins in detail. Besides the predominant enrichment of DNAH12 (top1), potential IDA interactors such as DNALI1 (top2), DNAH1 (top3), DNAH3 (top4), and DNAH2 (top5) were co-enriched in the candidate interacting dynein protein list (*Supplementary file 1a*). Immunoblotting confirmed the co-enrichment of DNAH12, DNALI1, and DNAH1 in the immunoprecipitated complex by the DNAH12 rabbit antibody in *Dnah12^+/+* mice but not in *Dnah12^-/-* mice, indicating specific interactions between DNAH12 and DNALI1, DNAH1 (*Figure 6E*). Reciprocal in vivo Co-IP using an anti-DNALI1 antibody further validated strong interactions among these three IDA components (*Figure 6F*).

Given the potential role of DNALI1 as an IDA assembly factor (*Wu et al., 2023*), we delved into the interaction details between DNAH12 and DNALI1. Employing Phyre2, we constructed the structural models of both proteins and further elucidated their interaction dynamics via DNAH12-DNALI1 docking using the HDOCK server. The results showed that the N-terminal Stem domain of DNAH12

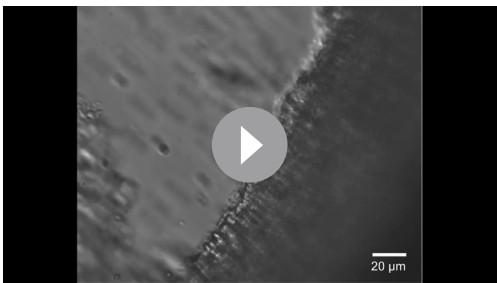

**Video 3.** Tracheal ciliary motility of *Dnah12^+/+* mice.
https://elifesciences.org/articles/100350/figures#video3

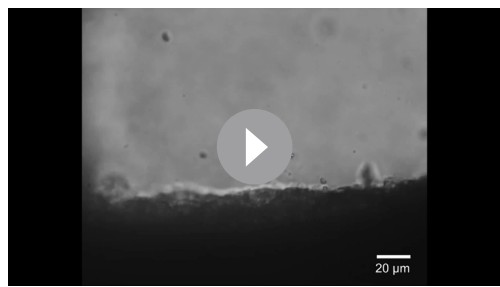

**Video 4.** Tracheal ciliary motility of *Dnah12^-/-* mice.
https://elifesciences.org/articles/100350/figures#video4

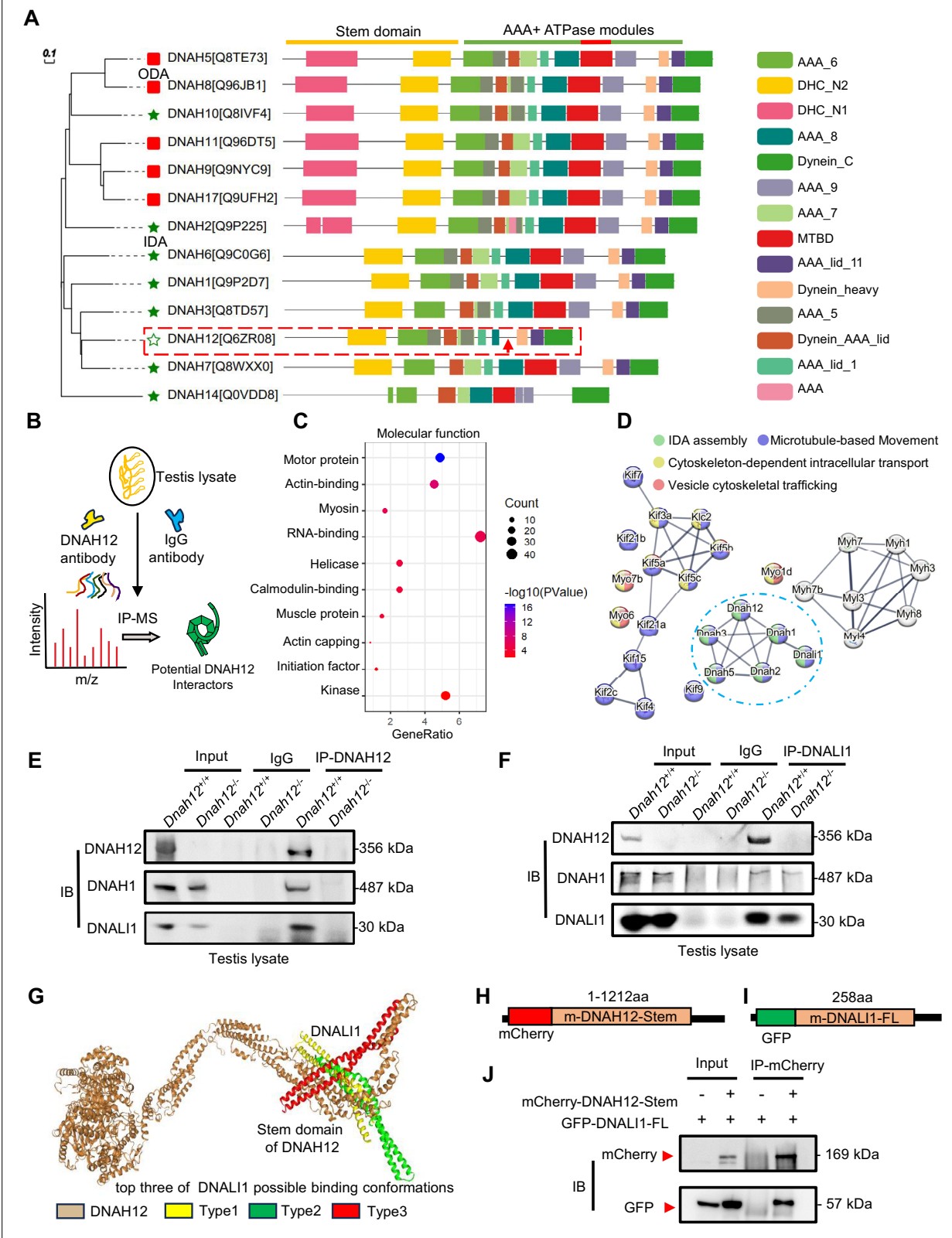

**Figure 6.** DNAH12 interacts with inner dynein arm (IDA) components DNALI1 and DNAH1. (**A**) Phylogenetic and protein domain analyses of the DNAH family. The dashed box indicates the shortest-length member, DNAH12, while the arrow points to the missing microtubule-binding domain (MTBD) region. (**B**) Schematic diagram showing the procedure of the immunoprecipitation/mass spectrometry (IP/MS) proteomics experiment. (**C**) GO term enrichment analysis of DNAH12 potential interactors. (**D**) Protein-protein interaction network of DNAH12 among the motor protein term. The blue circle

*Figure 6 continued on next page*

*Figure 6 continued*

marks the proteins involved in IDA assembly. (**E–F**) Co-immunoprecipitation (Co-IP) assays using DNAH12 antibody (**E**) or DNALI1 antibody (**F**) showed strong and specific interactions among DNAH12, DNALI1, and DNAH1. (**G**) Predicted DNAH12-DNALI1 interaction details by the HDOCK server. The top three of DNALI1 possible binding conformations were shown in types 1–3. (**H–J**) Construction of mouse-mCherry-DNAH12-Stem (**H**) and mouse-GFP-DNALI1-full length (FL) (**I**) plasmids, and the validation of the interaction between the Stem domain of DNAH12 and full-length of DNALI1 (**J**).

The online version of this article includes the following source data and figure supplement(s) for figure 6:

**Source data 1.** Labelled files for western blot in *Figure 6E, F and J*.

**Source data 2.** Original files for western blot in *Figure 6E, F and J*.

**Figure supplement 1.** DNAH12 lacks the microtubule-binding domain (MTBD) domain.

possibly interacts with DNALI1 in the top three best-fit conformations (*Figure 6G*). Considering DNAH12 is encoded by 3092 amino acids and their full-length protein interactions have been proved in vivo, we constructed the Stem domain (1–1212 amino acids) of DNAH12 and full-length DNALI1 plasmid, further in vitro Co-IP assays validated that the Stem domain of DNAH12 indeed mediates interactions with DNALI1 (*Figure 6H–J*). Together, these findings proved the specific interactions among DNAH12, DNALI1, and DNAH1. Given the significantly decreased protein levels of DNALI1 and DNAH1 observed in the sperm of *Dnah12^{-/-}* mice (*Figure 5B*), we proposed that the interactions involving DNAH12 (possibly restricted to the Stem domain of DNAH12), DNALI1, and DNAH1 are crucial for flagella development. As dynein proteins typically share similar domains, our findings also suggested that the Stem domain of DNAH family members may facilitate the interaction with DNALI1.

## Deletion of DNAH12 impairs the recruitment of IDA components during flagella development

To evaluate the effect of DNAH12 deletion on DNALI1 and DNAH1 during spermiogenesis, we examined the abundance of the DNAH12 interaction partners, DNALI1 and DNAH1, in *Dnah12^{+/+}* and *Dnah12^{-/-}* mouse testis lysate. The protein levels of DNALI1 and DNAH1 were decreased in *Dnah12^{-/-}* testes, while the level of ODA member DNAI1 was comparable (*Figure 7A*). We then stained DNALI1 or DNAH1 on testicular cell smear of *Dnah12^{+/+}* and *Dnah12^{-/-}* mice. Although the signals of DNALI1 or DNAH1 in round spermatids of *Dnah12^{-/-}* mice were comparable to those of *Dnah12^{+/+}* mice (*Figure 7B and C*), the recruitment of DNALI1 and DNAH1 to sperm flagella was severely impaired in elongated spermatids of *Dnah12^{-/-}* mice (*Figure 7D and E*). Considering that the deficiency of DNAH12 also caused scarcely visible immunofluorescence signals of DNALI1 and DNAH1 in sperm flagella of humans and mice (*Figures 3F, G, 5E and F*), we proposed that the interaction among DNAH12, DNALI1, and DNAH1 is conserved in mammals, and is required for flagellar development during spermiogenesis. Since the deficiency of DNAH12 does not cause PCD symptoms in humans and mice, we hypothesized that flagella and cilia may exhibit distinct features in the process of dynein assembly. We thus checked the abundance of DNALI1 and DNAH1 in the oviduct and trachea, the results showed that DNALI1 and DNAH1 were comparable in the *Dnah12^{+/+}* and *Dnah12^{-/-}* groups (*Figure 7F and G*). Furthermore, no discernible disparities were observed in the intensity and localization of DNALI1 and DNAH1 signals within the oviduct and trachea (*Figure 7H–K*). In summary, these results underscore the specific involvement of DNAH12 in flagellar development, rather than in cilial development. Moreover, we supposed that the assembly of IDA components differs between flagella and cilia.

## DNAH12 is prerequisite for the recruitment of DNALI1 and DNAH1 to sperm flagella

In patients and mice with DNALI1 deficiency, flagellar DNAH1 but not DNAH12 signals were absent (*Wu et al., 2023*). To unravel the recruitment relationships among DNAH12, DNALI1, and DNAH1 during sperm flagellar formation, we conducted staining of DNALI1 and DNAH12 on sperm flagella obtained from the infertile asthenoteratozoospermic male P13 with a homozygous *DNAH1* mutation (c.11275C>T) of a family (*Figure 7L*, *Figure 7—figure supplement 1A-B*), wherein DNAH1 signals were absent along the flagella (*Figure 7M*). In comparison to those in the control, the DNALI1 signals were absent, while no discernible differences in DNAH12 signals were observed in the flagella of the patient with the *DNAH1* mutation (*Figure 7N and O*). These findings suggested that the recruitment

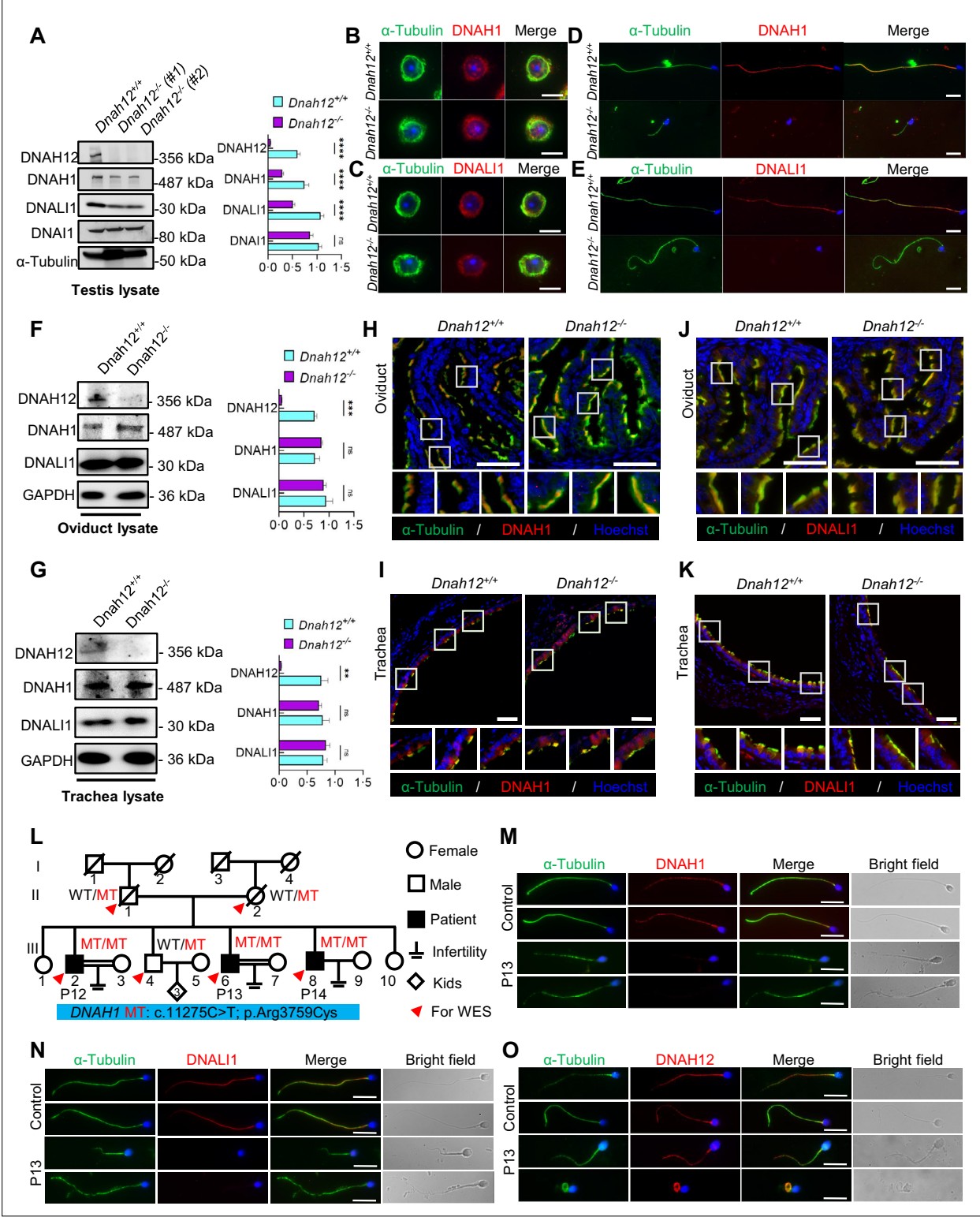

**Figure 7.** DNAH12 facilitates the recruitment of DNALI1 and DNAH1 to flagella, but not cilia. (**A**) Immunoblotting of testis lysate from *Dnah12*<sup>+/+</sup>, *DNAH12*<sup>+/-</sup> and *Dnah12*<sup>-/-</sup> mice using DNAH12, DNAH1, and DNALI1 antibodies. DNAI1 (a component of ODAs) and α-Tubulin were used as the loading controls. The relative band grayscales of proteins to α-Tubulin in testis lysate from *Dnah12*<sup>+/+</sup>, and *Dnah12*<sup>-/-</sup> mice were shown on the right (n=5 independent experiments). Data are presented as mean ± SEM; ns indicates no significant difference;****p<0.0001. (**B–C**) Immunofluorescence assays of α-Tubulin and DNAH1 (**B**), or DNALI1 (**C**) antibodies on round spermatids from adult *Dnah12*<sup>+/-</sup> and *Dnah12*<sup>-/-</sup> mice. Scale bars, 10 μm. (**D–E**) Immunofluorescence assays of α-Tubulin and DNAH1 (**D**), or DNALI1 (**E**) antibodies on elongated spermatids from adult *Dnah12*<sup>+/-</sup> and *Dnah12*<sup>-/-</sup>

*Figure 7 continued on next page*

Figure 7 continued

mice. Scale bars, 10 µm. (**F–G**) Immunoblotting of oviduct (**F**) or trachea (**G**) lysate from *Dnah12*$^{+/+}$, *DNAH12*$^{+/-}$ and *Dnah12*$^{-/-}$ mice using DNAH12, DNAH1, and DNALI1 antibodies. GAPDH was used as the loading control. The relative band grayscales of proteins to GAPDH in oviduct lysate (**F**) or trachea lysate (**G**) from *Dnah12*$^{+/+}$, and *Dnah12*$^{-/-}$ mice were shown on the right (n=4 independent experiments). Data are presented as mean ± SEM; ns indicates no significant difference; ***$p<0.001$. (**H–I**) Immunofluorescence assays of the oviduct (**H**) or trachea (**I**) sections with α-Tubulin and DNAH1 antibodies. Scale bars, 50 µm. (**J–K**) Immunofluorescence assays of the oviduct (**J**) or trachea (**K**) sections with α-Tubulin and DNALI1 antibodies. Scale bars, 50 µm. (**L**) Pedigree of a family with three infertile males, P12 (III-2), P13 (III:6), and P14 (III:8); MT: the *DNAH1* mutation c.11275C>T; p.Arg3759Cys. (**M–O**) Representative images of spermatozoa from a fertile control and P13 co-stained by α-Tubulin antibody and DNAH1 (**M**), DNALI1(**N**), or DNAH12 (**O**) antibodies, respectively. Scale bars,10 µm.

The online version of this article includes the following source data and figure supplement(s) for figure 7:

**Source data 1.** Labelled files for western blot in *Figure 7*.

**Source data 2.** Original files for western blot in *Figure 7*.

**Figure supplement 1.** Flowchart for analyses of the whole-exome sequencing of a Pakistani family and validation of *DNAH1* mutation through Sanger sequencing.

of DNAH12 to flagella is independent of DNAH1, and DNAH12 is the first protein to be recruited to flagella among the three proteins.

## DNAH12 deficiency causes the decrease of RS head-associated proteins

In addition to the enrichment of dynein proteins, we noticed that clusters of RS head-related proteins such as RSPH1, RSPH9, or DNAJB13 were enriched in the testicular interactome of DNAH12. Considering that deficiency in RSPH1 (*Lin et al., 2014*), RSPH9 (*Zhu et al., 2019*), or DNAJB13 (*Liu et al., 2022*) were associated with CP loss phenotype. We then explored their interaction with DNAH12 in the testis lysate, the results proved the interactions of these proteins with DNAH12 in the testis (*Figure 8A*). Further analyses by AlphaFold3 showed that these proteins may interacts with DNAH12 through the Stem domain (*Figure 8B*). The expression of RSPH1, RSPH9, and DNAJB13 showed an obvious decrease in testis lysate of *Dnah12*$^{-/-}$ mice (*Figure 8C*). Additionally, immunofluorescence assays revealed the loss of signals for RSPH1, a marker of RS head, in *Dnah12*$^{-/-}$ sperm flagella (*Figure 8D*). Together, these results indicate that deficiency of DNAH12 lead to the impairment of radial spoke head, which could contribute to CP loss in flagella of *Dnah12*$^{-/-}$ mice.

## ICSI may overcome the infertility caused by DNAH12 deficiency

Since the DNAH12 mutated patients and the *Dnah12*$^{-/-}$ mice have similar deficiency, to evaluate the prospects of whether the asthenoteratozoospermia patients caused by *DNAH12* variants could be treated with assisted reproductive technology, spermatozoa from D*nah12*$^{+/+}$ and *Dnah12*$^{-/-}$ mice were used for ICSI. Promisingly, the percentages of 2-cell and blastocyst-stage embryos in *Dnah12*$^{-/-}$ group showed no difference compared to those of *Dnah12*$^{+/+}$ group (*Figure 8E–G*), indicating that ICSI could also be an effective treatment for infertile men carrying *DNAH12* variants.

Taken together, we suggest that DNAH12 interacts with DNALI1 and DNAH1, and this interaction is a prerequisite for the proper localization of DNALI1 and DNAH1 to sperm flagella, which is essential for the intact IDAs and CP structures in both humans and mice. Additionally, DNALI1 and DNAH1 exhibit mutual dependence for their localization to sperm flagella. Deficiency of DNAH12 leads to the failure of DNALI1 and DNAH1 recruitment to sperm flagella, resulting in abnormal sperm morphology characterized by compromised axoneme organization, causing male infertility (*Figure 8H*). Moreover, our results indicate that the assembly of IDA components occurs intricately, possibly not synchronously in time and space.

## Discussion

In this study, we identified six novel bi-allelic mutations in *DNAH12* that cause the loss of DNAH12 proteins in asthenoteratozoospermia patients. Spermatozoa from these patients exhibited abnormal morphology and compromised axonemal defects, including a high percentage of impaired IDAs and loss of CP. Two mouse models (*Dnah12*$^{-/-}$ and *Dnah12*$^{mut/mut}$) were generated, recapitulating the abnormal flagellum morphology and structure observed in patients, thereby confirming the pathogenic

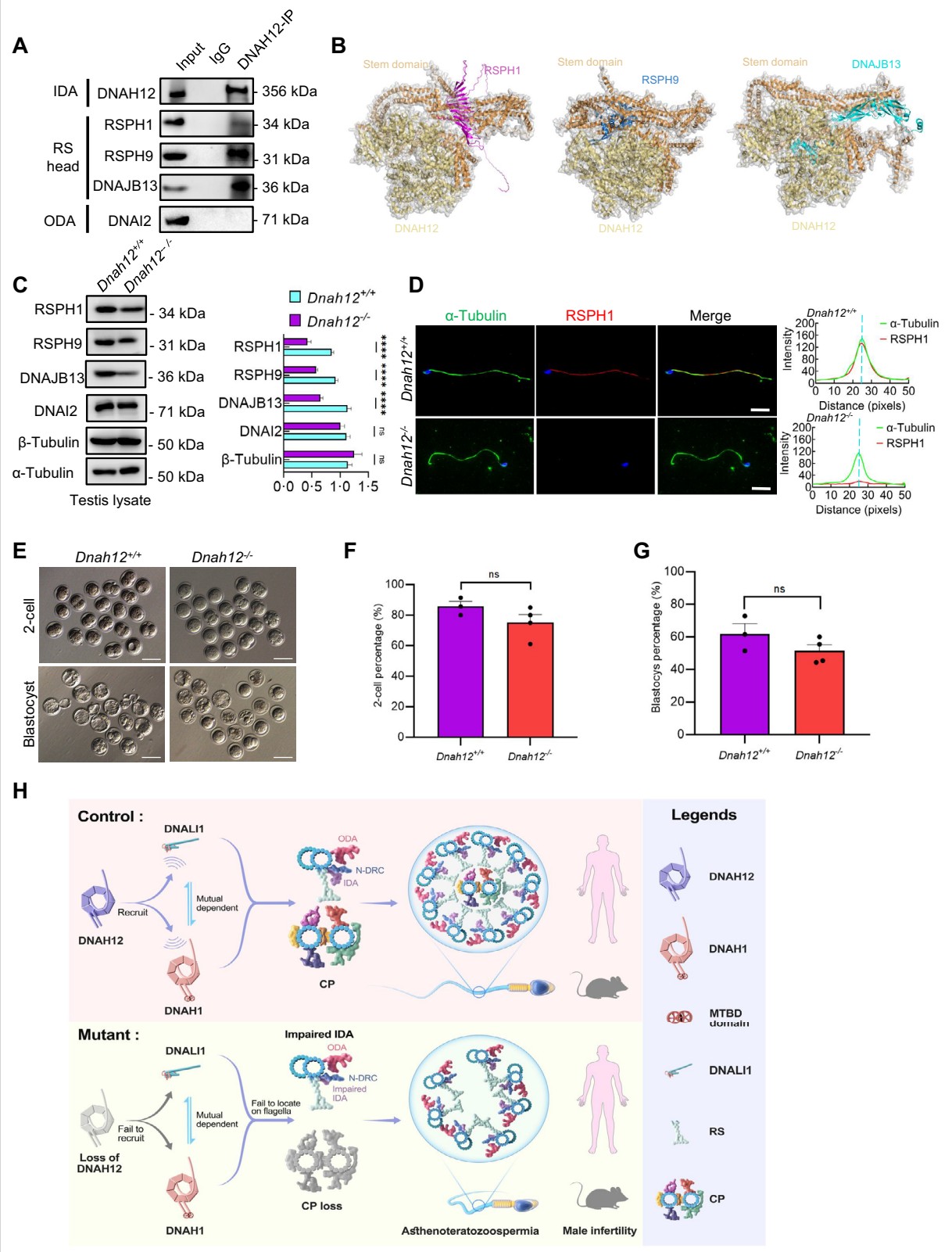

**Figure 8.** DNAH12 may interact with RS head-associated proteins to regulate CP stability. (**A**) Co-immunoprecipitation (Co-IP) assays of the potential interactions of DNAH12 and RS head proteins RSPH1, RPSH9, or DNAJB13. DNAI2 antibody was used as a negative control. (**B**) Interaction analyses of DNAH12 and RS head proteins RSPH1, RPSH9, or DNAJB13 by AlphaFold3. (**C**) Immunoblotting of testis lysate from *Dnah12⁺/⁺* and *Dnah12⁻/⁻* mice using RSPH1, RPSH9, DNAJB13 antibodies. DNAI2 (a component of ODAs), α-Tubulin, and β-Tubulin were used as the loading controls. The relative band

*Figure 8 continued on next page*

*Figure 8 continued*

grayscales of proteins to α-Tubulin in testis lysate from *Dnah12⁺/⁺*, and *Dnah12⁻/⁻* mice were shown on the right (n=4 independent experiments). Data are presented as mean ± SEM; ns indicates no significant difference;****p<0.0001.; (**D**) Immunofluorescence assays of α-Tubulin and RSPH1 antibodies on sperm collected from *Dnah12⁺/⁻* and *Dnah12⁻/⁻* mice caudal epididymis. The fluorescent signal intensity profiles were shown on the right. Scale bars, 10 µm. (**E**) Representative two-cell embryos and blastocysts of *Dnah12⁺/⁺*, and *Dnah12⁻/* male mice after intracytoplasmic sperm injection. Scale bar, 100 µm. (**F**) Percentages of two-cell-stage embryos of *Dnah12⁺/⁺*(n=3 mice), and *Dnah12⁻/⁻* male mice (n=4 mice) after intracytoplasmic sperm injection. Data are presented as mean ± SEM; ns indicates no significant difference. (**G**) Percentages of blastocyst-stage embryos of *Dnah12⁺/⁺*(n=3 mice), and *Dnah12⁻/⁻* males (n=4 mice) after intracytoplasmic sperm injection. Data are presented as mean ± SEM; ns indicates no significant difference. (**H**) Schematic diagram showing the proposed function of DNAH12 in sperm flagellar development in humans and mice. DNAH12 is essential for recruiting DNALI1 and DNAH1 to sperm flagella and maintaining the proper axonemal arrangement, especially IDAs and CP structures. Loss of DNAH12 causes the failure of DNALI1 and DNAH1 to be recruited to sperm flagella and results in abnormal sperm morphology with compromised axoneme organization, causing male infertility. ODA, outer dynein arm; IDA, inner dynein arm; N-DRC, nexin-dynein regulatory complex; CP, central pair of microtubules; MTBD, microtubule-binding domain; RS, radial spoke.

The online version of this article includes the following source data for figure 8:

**Source data 1.** Labelled files for western blot in *Figure 8A and C*.

**Source data 2.** Original files for western blot in *Figure 8A and C*.

role of *DNAH12* mutations in asthenoteratozoospermia. Through testicular Co-IP and mass spectrometry proteomics, we screened and validated that DNAH12 interacts with IDA components like DNALI1 and DNAH1 and is indispensable for recruiting DNALI1 and DNAH1 to sperm flagella, highlighting the crucial role of DNAH12 in sperm flagellar development. Notably, deficiency of DNAH12 does not manifest PCD symptoms in humans and mice, underscoring the unique role of DNAH12 in flagellar development. Collectively, our study not only unveils the functional significance of the previously unexplored gene *DNAH12* but also elucidates the pathogenic link between *DNAH12* mutations and asthenoteratozoospermia.

DNAH12 and other dynein arm proteins share the N-terminal Stem domain, which is crucial for structure and accessory subunit binding (*Kollmar, 2016*), and the motor domain which encompasses ~6 AAA + ATPase modules forming the ring-shaped oligomeric complexes responsible for force production (*Canty et al., 2021*). Five of the identified *DNAH12* mutations, primarily in the Stem domain (critical for DNALI1 interaction), result in loss-of-function effects, indicating a weak tolerance property of the Stem domain. Similar weak tolerance features were observed in mutations of other dynein arm components like *DNAH1* and *DNAH2* (*Sha et al., 2017*; *Hwang et al., 2021*). The profound impact of these variants, particularly missense mutations, underscores delicate features of the Stem domain and DNAH family members. Notably, DNAH12 stands out as the shortest in sequence length and the only one lacking the essential MTBD region within the DNAH family (*Lacey et al., 2019*). MTBD, a 15-nanometer coiled-coil stalk structure, crucially supports ATP-sensitive microtubule binding properties and determines dynein motility direction (*Carter et al., 2008*). Our investigation revealed that DNAH12 interacts with other IDA components, specifically DNALI1 and DNAH1. Despite the mutual dependence between DNAH1 and DNALI1 for their flagellar localization, DNAH12 is prerequisite for the proper assembly of DNALI1 and DNAH1 onto sperm flagella in both humans and mice. Notably, *Dnah12⁻/⁻* mice exhibit more severe phenotypes than *Dnali1* and *Dnah1* knockout mice (*Khan et al., 2021*; *Wu et al., 2023*; *Yap et al., 2023*), including a more severe reduced sperm count (~2% of normal values), this observation further supports our perspective.

Deficiency in DNAH12 presented sperm ultrastructural defects like CP loss and impaired IDAs indicating essential roles of DNAH12 in the maintenance of axonemal integrity and arrangement. These results indicate essential roles of DNAH12 in the maintenance of axonemal integrity and arrangement. Noteworthy, deficiency of DNAH12 potential partner DNAH1, a well-known asthenoteratozoospermia-related gene, presents quite similar defects with DNAH12, indicating DNAH12 and DNAH1 possibly exhibit synergic functions in axonemal proper arrangement (*Ben Khelifa et al., 2014*; *Wambergue et al., 2016*; *Wang et al., 2023*). In addition to DNAH12 and DNAH1, deficiency in DNAH2 or DNAH6 also leads to high proportions of CP loss and disorganized axonemal structures (*Li et al., 2019*; *Hwang et al., 2021*; *Gao et al., 2021*). However, DNALI1, DNAH7, or DNAH10 deficiency results in no obvious CP loss but impaired IDA integrity (*Gao et al., 2022a*; *Li et al., 2022*; *Wei et al., 2021*; *Wu et al., 2023*). These studies indicate that specific IDA components, like DNAH12 and DNAH1, are required for the maintenance of CP integrity. Moreover, discrepancies observed in the phenotypes

resulting from deficiencies in specific IDAs suggest that multiple dynein proteins are produced to support the intricate organization of sperm flagella. Additionally, these proteins may exhibit distinct characteristics and perform unique functions to some extent. Noteworthy, deficiency in RS head proteins RSPH1, RSPH9, or DNAJB13 were related to axonemal CP loss phenotype (*Lin et al., 2014*; *Liu et al., 2022*; *Zou et al., 2020*), indicating that DNAH12, and other IDAs, possibly regulate the stability of CP through the cooperation with RS head proteins. Notably, while the adoption of Alpha-Fold3 and HDOCK contributes to a clearer model of protein structures and their interactions, further techniques like cryo-EM may facilitate a deeper understanding of these proteins at higher resolution. Elucidating the dependencies among the assembly of sperm components could enhance our under-standing of the structural basis of sperm motility and human fertility. Additionally, while an enrichment of IDA and RS proteins, but not nexin-dynein regulatory complex (N-DRC) proteins, was observed in the DNAH12 interactome, further exploration is needed to determine whether DNAH12 may indi-rectly affect the stability and functionality of the N-DRC structure.

The localization of DNAH12 presents dynamic changes during spermiogenesis. DNAH12 is initially expressed in the cytoplasm of the round spermatid, while it's enriched in the manchette structure of the elongating spermatid and the tail of the elongated spermatid. The manchette and the sperm tail axoneme are two microtubular platforms that transport proteins to the developing head and tail (*Lehti and Sironen, 2016*). Manchette is a transient skirt-like structure and functions as a scaffold for morphological remodeling, contributing to the formation of the sperm head, neck, and tail (*Hu et al., 2023b*). Here, we found that DNAH12 deficiency results in abnormal manchette structure and compromised sperm head and tail morphology. Additionally, in sperm of *Dnah12⁻/⁻* males, several CP complex proteins like SPAG6, and SPEF2 are not properly assembled and the axoneme structure of the tail was severely disorganized, indicating dynein components are responsible for proper axonemal components organization. Noteworthy, DNALI1, a potential DNAH12 interacting partner showed similar localization patterns to DNAH12, a study presented that DNALI1 recruits and stabilizes manch-ette component PACRG, and DNALI1 deficiency causes impaired sperm flagellar assembly (*Yap et al., 2023*). These data hint that dynein proteins like DNAH12 and DNALI, possibly work together in the manchette organization as well as sperm flagellar components assembly during sperm development.

DNAH12 was also detected in the lungs, tracheas, and female oviducts while no cilia morphology defects in these tissues or PCD symptoms were observed in both two mice models. Besides, none of the patients carrying bi-allelic *DNAH12* mutations claimed symptoms of PCD via visits except for the reproductive problems. *Dnah12⁻/⁻* female mice showed normal development and fertility and the mother (III:2) of patients from family 2, carrying homozygous *DNAH12* mutation M2 had 6 offspring. Intriguingly, we found that the localization and abundance of DNALI1 and DNAH1 in oviductal or tracheal cilia were not affected after *Dnah12* deletion, indicating that DNAH12 works differently in flagella and cilia. Deficiency of specific DNAH members like DNAH6 (*Li et al., 2016*), or DNAH7 (*Wei et al., 2021*), but not DNAH2 (*Hwang et al., 2021*), results in PCD symptoms. Hence, DNAH family members possibly present disparate processing of axonemal biogenesis and organization through evolution in flagella and cilia, further explorations could contribute to a more comprehensive under-standing of the differences between flagella and cilia, and personalized clinical diagnosis of related diseases like male/female infertility and/or PCD (*Zhang et al., 2022*).

Previously, two genetic screening studies mentioned unverified compound heterozygous muta-tions and a homozygous missense mutation in *DNAH12*, predicting their association with patients' infertility. However, limited genetic evidence was provided, and the pathogenesis remained unclear (*Li et al., 2021*; *Oud et al., 2021*). Through investigations involving patients with *DNAH12* vari-ants and mutant mouse models, we provide direct genetic and molecular evidence that variants in *DNAH12* are genetic causes of asthenoteratozoospermia, and DNAH12 along with DNAH1, DNALI1 is indispensable for sperm flagellar biogenesis. These data strongly support DNAH12 as a valuable marker in genetic counseling and diagnosis of male infertility.

ICSI is a widely used assisted reproductive technology in treating infertile men with asthenoter-atozoospermia. Deficiency in most dynein proteins leads to asthenoteratozoospermia with severely compromised sperm motility and morphology. Previous studies have shown that patients carrying mutations in *DNAH17 Whitfield et al., 2019* have poor ICSI outcomes while patients with *DNAH1*(59), *DNAH2* (*Li et al., 2019*; *Gao et al., 2021*), *DNAH3* (*Meng et al., 2024*), *DNAH7* (*Wei et al., 2021*), or *DNAH8* (*Weng et al., 2021*) mutations have achieved favorable clinical pregnancy. Thus, the

comparative study of asthenoteratozoospermia-related genes on their ICSI outcomes will be beneficial to developing individualized treatment recommendations in clinical practice. In our study, the successful ICSI outcomes of *Dnah12*<sup>-/-</sup> mice suggest that ICSI could be used as a potential treatment for infertile men carrying *Dnah12* mutations. It's meaningful to evaluate the prognosis for ICSI treatments in patients carrying *DNAH12* variants in a large population.

In conclusion, our study functionally and mechanistically sheds light on a special and poorly characterized protein DNAH12, during the intricate process of sperm flagellar biogenesis. These results pave the way for an enhanced understanding of the roles of dynein proteins in sperm flagellar development and their functional difference in flagella and cilia, offering a diagnostic target for genetic counseling and future individualized treatment of male infertility in clinical practice.

## Materials and methods

### Human subjects

Informed consent and peripheral blood were obtained from the participants and the patients' information was registered in the Human Reproductive Disease Resource Bank database. Sperm analyses were performed according to the WHO guidelines (*WHO, 2021*). During their medical consultation, all patients answered a questionnaire regarding clinical presentations. This study was approved by the institutional ethics committee of the University of Science and Technology of China (USTC) with the approval number 2019-KY-168.

### Whole exome sequencing (WES) and bioinformatics analyses

Whole-exome capture and sequencing were performed by capturing exons from genomic DNAs of peripheral blood cells, using the SureSelect XT 50 Mb Exon Capture Kit (Agilent, California, USA), and then high-throughput sequencing was conducted on the HiSeq 2000 platform (Illumina, San Diego, CA, USA) following instructions. Variants were called using the Genome Analysis Toolkit Haplotype-Caller and annotated through ANNOVAR (*Wang et al., 2010*). Variants within exons or exon-intron boundaries were retained and filtered through the criteria we previously described (*Yin et al., 2019*). Bcftools was applied to detect runs of homozygosity (*Narasimhan et al., 2016*). Runs of homozygosity over 1.5 Mb were used to calculate the inbreeding coefficients. The genotypes were validated using Sanger sequencing validation. Additional primers are listed in ***Supplementary file 1, table 1b1***. For phylogenetic analysis, different DNAH12 protein sequences were extracted from the UniProt database, and the sequence alignment and phylogenetic tree construction were generated using MEGA11 (*Tamura et al., 2021*). Newick format tree files were extracted and processed in the Evolview v2 tool (*He et al., 2016*). For gene family and conserved domain analyses, the batch sequence search tool in Pfam was used to analyze features of proteins and further visualized with the assistance of Evolview v2 tool and TBtools (*Chen et al., 2020*; *Mistry et al., 2021*). For protein structure analysis, the structure models were predicted and constructed with the Phyre2 tool (*Kelley et al., 2015*), interactions were analyzed by the HDOCK (http://hdock.phys.hust.edu.cn/) and Alphafold3 (https://golgi.sandbox.google.com/). The predicted protein structures and possible interactions were further explored and visualized by UCSF Chimera (Version: 1.15, California, USA) according to the manual instruction (*Pettersen et al., 2004*).

### Scanning electron microscopy (SEM) and transmission electron microscopy (TEM)

Fresh semen samples were obtained and centrifuged at 900 g for 3 min. Spermatozoa were washed with phosphate buffer saline (PBS) three times. Then the sediment was fixed in 0.1 M phosphate buffer (pH 7.4) containing 4% paraformaldehyde, 8% glutaraldehyde, and 0.2% picric acid at 4 °C for at least 8 hr. SEM and TEM were then conducted as we previously described (*Ma et al., 2021*).

### Generation and verification of the polyclonal anti-DNAH12 antibody

The characteristics of the DNAH12 including its region immunogenicity, hydrophilicity, surface leakage groups, and sequence homology were analyzed and the 1–200 amino acids sequence of mouse DNAH12 protein (UniProt accession no. Q69Z23) was regarded as the ideal antigen for immunization. Subsequently, the DNAH12 polyclonal antibody was generated in rabbits and rats using the

mentioned peptide by Dia-An Biotech, Inc, in Wuhan, China. The specificity of antibodies was further validated using *Dnah12$^{+/+}$* and *Dnah12$^{-/-}$* mice in this study.

## Immunoblotting

Protein lysate was prepared as we previously described (*Ma et al., 2022*; *Zhang et al., 2021*). Tissues or cells were lysed using lysis buffer (50 mM tris (pH 7.5), 150 mM NaCl, 5 mMEDTA, and 0.1% NP-40). For human sperm cells, proteins were extracted using the TRIzol reagent. The protein sediment was washed with 95% ethanol containing 0.3M guanidine hydrochloride, dissolved in 1% SDS. Then the protein lysate was boiled for 10 min, and diluted in SDS-sample buffer (62.5 mM TRIS, 2% (w/w) SDS, 10% (w/v) glycerol, and 5% 2-mercaptoethanol). Whereafter the proteins were separated by SDS-PAGE and transferred onto nitrocellulose blotting membranes with a pore size of 0.45 µm (GE Healthcare, 10600002). The membranes were blocked with TBS-Tween buffer (50 mM Tris, 150 mM NaCl, and 0.5% Tween-20, pH 7.4) containing 5% skim milk for 30 min to reduce unspecific binding and then incubated with primary antibodies diluted in TBST buffer containing 5% skim milk at 4 °C overnight. Incubation with secondary antibodies for 1.5 hr was followed by chemiluminescence development system (GE Healthcare, Pittsburgh, USA). Antibodies details are listed in *Supplementary file 1c and d*.

## Immunofluorescence (IF) staining

Human semen smear slides were prepared following the guidelines of WHO (*WHO, 2021*). Mouse sperm smear slide preparation and immunofluorescence staining were conducted as we described (*Dil et al., 2023*). For tissue IF staining, tissues were collected and fixed in 4% paraformaldehyde (PFA), the 5µm tissue sections were deparaffinized in xylene, rehydrated in a graded series of ethanol (100%, 95%, 90%, 80%, 70%, and 50% ethanol) and sterile water, and permeabilized with 0.2% Triton X-100 in PBS for 30min. Antigen retrieval was then conducted by boiling slides in a microwave for 30 min in 10 mM citrate buffer (pH 6.0). The other steps are consistent with sperm IF staining. Images of spermatozoa were captured using a Nikon ECLIPSE 80i microscope equipped with a charge-coupled device (Hamamatsu Photonics, Hamamatsu, Japan). The fluorescent signal intensity profiles were depited using Fiji software (https://imagej.net/software/fiji/). Information of antibodies is available in *Supplementary file 1c and d*.

## Animal models and fertility test

Two mouse models (*Dnah12$^{-/-}$* and *Dnah12$^{mut/mut}$*) were generated using CRISPR/Cas9 Technology, respectively (*Wang et al., 2013*). Briefly, to generate the *Dnah12$^{-/-}$* mouse model, guide RNAs (gRNAs) targeting exon 5 of *Dnah12*, a key exon shared by all its transcripts, were transcribed in vitro (Thermo Fisher, CA, USA). Then, the gRNAs and the Cas9 protein were electroporated into C57BL/6 mouse zygotes. The zygotes were introduced into Institute of Cancer Research (ICR) female mice, and the genotypes of the pups were checked using Polymerase Chain Reaction (PCR) with designed primers, the mice with the frameshift mutation c.386_389 del that may lead to paralyzed DNAH12 function were kept for further study. Similar conductions were performed targeting exon 18 of *Dnah12* to generate the *Dnah12$^{mut/mut}$* mice. The genotyping primers are listed in *Supplementary file 1b*. For male fertility test, each 8–10 wk male mouse was caged with wild-type female mice; for female fertility test, each 8–10 wk female mouse was caged with one 8–10 wk *Dnah12$^{+/-}$* male mouse, the number of offspring was recorded.

## Reverse-transcription PCR and real-time quantitative PCR

For reverse-transcription PCR (RT-PCR), total RNA was extracted from mouse tissues using TRIzol reagent (TAKARA, Japan). Whereafter 1 µg of RNA was reverse-transcribed using the PrimeScript RT Reagent Kit (TAKARA, Japan). The cDNA was diluted to a comparable concentration and then used for RT–PCR. All amplified DNA fragments were visualized by 2% agarose gel electrophoresis containing GelRed (APExBIO, Houston, TX, USA). For real-time quantitative PCR (qPCR), the experiments were performed by using Hieff UNICON Universal Blue qPCR SYBR Green Master Mix reagent (Yeasen Biotechnology, Shanghai, China) on CFX96 Real-Time System (Bio-Rad, CA, USA) according to the manufacturer's guidelines. Details of the primers were listed in *Supplementary file 1b* .

## Histological analysis

As previously described (*Yin et al., 2019*), after sacrificing mice, the testes, epididymides, tracheas, and oviducts were harvested and fixed in Bouin's fixative solution (Sigma-Aldrich, Missouri, USA) for hematoxylin and eosin (H&E) staining or 4% PFA fixative solution for immunofluorescence or TUNEL assay. The pictures were captured using a microscope equipped with a charge-coupled device camera (Nikon, Tokyo, Japan).

## Sperm counts, motility, and morphology analysis

As previously described (*Xu et al., 2022*), H&E staining or Papanicolaou staining kit (Solabio, Beijing, China) was used for the evaluation of sperm morphology of patients. For mouse sperm parameters analysis, 8–10 wk-old mice were sacrificed by cervical dislocation, the sperm number per epididymis was then assessed by cutting the epididymis into pieces in PBS and incubating them for 30 min at 37 °C to release the sperm, then counting with a hemocytometer. Sperm motility were assessed by computer-aided sperm analysis (CASA) as we described (*Ma et al., 2023*). Sperm morphology was assessed on spermatozoa collected from epididymal cauda after H&E staining.

## Testicular cell smear slides and seminiferous tubule squash slides preparation

To prepare testicular cell smears, mice were sacrificed and testes were then detached with one time of PBS washing. Then testes were peeled off the testicular capsule, torn into small pieces, and then cut into tiny pieces in PBS. Finally, the cell suspension was filtered through a 200-mesh cell strainer, the filtered cell suspensions were added 10% FBS before spread onto glass slides. For testicular seminiferous tubule squash slide preparation, the testes were dissected and carefully isolated into single testicular seminiferous tubules in PBS dishes. After 10 min of prefixation in fix/lysis solution (2% PFA diluted in PBS), we then applied 100 μL fix/lysis solution onto poly-L-lysine coated glass slides and carefully teased apart approximately 20 seminiferous tubules with the assistance of two sterile forceps on the slides. After that, coverslips were applied using slide forceps and pressed with the heel of the palm for 10 s, then the glass slides were frozen in a small container with liquid $N_2$ immediately till the liquid ceases to bubble. The slides were saved in the –80 °C fridge keeping the coverslip on the slides or removing coverslips for immunofluorescence staining.

## TUNEL assay

As previously described (*Xu et al., 2022*), In Situ Cell Death Detection Kit, Fluorescein (Sigma–Aldrich, Missouri, USA) was applied according to the manufacturer's protocol. Images were captured using a Nikon Eclipse 80i microscope equipped with Hamamatsu C4742-80 digital camera and analyzed using the NIS-Element Microscope imaging software (Nikon, Tokyo, Japan).

## Cell culture and transfection

HEK293T (Cat# CRL-3216, RRID:CVCL_0063, ATCC, VA, USA) cells were authenticated by short tandem repeat (STR) profiling, and tested negative for mycoplasma contamination, No other cell lines from the list of commonly misidentified cell lines maintained by the International Cell Line Authentication Committee were used. For the cell culture and transfection, Aas previously described (*Ma et al., 2022*), HEK293T cellsHEK293T (ATCC, VA, USA) cells were cultured to 70%–80% density in 12-well plates, followed by transfected with plasmids expressing mCherry-tagged mouse DNAH12 and GFP-tagged mouse DNALI1 using Lipofectamine 3000 (Thermo Fisher, CA, USA) according to the manufacturer's instructions. Cells were harvested for in-vitro IP and immunoblotting the next day after transfection. The sequences of the primers for plasmid construction are listed in *Supplementary file 1d*.

## Co-immunoprecipitation (Co-IP)

Tissue extracts were prepared using lysis buffer supplemented with PMSF, Protease Inhibitor Cocktail (Solarbio, Beijing, China) on ice using a glass homogenizer. Lysate was then centrifuged at 12,000 g at 4 °C for 10 min, and the supernatant was divided into equal aliquots. Each aliquot was incubated with 1.5 μg of target antibody or rabbit IgG nonspecific antibody. After incubation at 4 °C overnight with rotation, beads were washed five times with lysis buffer. Finally, the beads were resuspended in

SDS-sample buffer and boiled at 100 °C for 10 min. The samples were either analyzed by immunoblotting immediately or stored at –80 °C. For Co-IP in cultured cells, HEK293T cells were plated in six-well plates, followed by transfection using Lipofectamine 3000 reagents. MG132 (5 μM) was added into the cultured cells 6 hr before cell harvesting. Cells were washed with PBS for 10 min, cells were then collected and lysed in the same lysis buffer used for Co-IP from testicular samples. The cell lysate was then incubated with prewashed Protein A/G Magnetic Beads B23202 (Bimake, Texas, USA) and primary antibodies at 4 °C overnight, followed by washing and boiling as described above for the Co-IP protocols for testicular samples.

## Mass spectrometry proteomics

To identify potential partners that interact with DNAH12, we performed IP of mouse testes lysate. Tissue extracts were prepared using lysis buffer supplemented with PMSF, Protease Inhibitor Cocktail (Solarbio, Beijing, China) with the same operation of the Co-IP protocol. Beads were washed three times with lysis buffer and three times with 50 mM $NH_4HCO_3$, the pull-down proteins were digested with 0.2 μg Trypsin overnight and stopped by 5% Formic acid, (Thermo Fisher, CA, USA). The peptide mixture was then desalted, dried with a SpeedVac, and resuspended in 0.1% Formic acid, followed by the MS analysis using timsTOF Pro 2 mass spectrometer (Bruker, Massachusetts, USA). The proteomics data are available on the ProteomeXchange Consortium via the iProX partner repository (*Chen et al., 2022*), with the dataset identifier PXD051681.

## Intracytoplasmic sperm injection (ICSI)

3.5–4 wk-old female mice were given an intraperitoneal injection of 5 IU Pregnant Mare Serum Gonadotropin (PMSG, Ningbo Sansheng, China). After 48 hr, a second intraperitoneal injection of 5 IU human Chorionic Gonadotropin (hCG, Ningbo Sansheng, China) was administered. The mice were euthanized by cervical dislocation after 15 hr post-hCG injection. The oviducts were excised and the ampullary regions were opened to release the cumulus-oocyte complexes (COCs). The cumulus cells surrounding the oocytes were digested using hyaluronidase, and the morphologically round MII stage oocytes were selected and washed three times in M2 medium (Sigma-Aldrich, Missouri, USA) before being set aside for use. Male mice are euthanized by cervical dislocation, the epididymides are excised and gently punctured with a syringe, allowing the spermatozoa to flow out. The spermatozoa were then placed in the human tubal fluid (HTF) medium (EasyCheck, Nanjing Aibei Biotechnology, China) and then frozen and thawed repeatedly to remove sperm tails, the single sperm head was microinjected into an MII oocyte and then cultured in potassium simplex optimized medium (KSOM, EasyCheck, Nanjing Aibei Biotechnology, China) at 37 °C under 5% $CO_2$.

## Statistics analysis

Statistical analyses were performed using GraphPad Prism 8.0 (GraphPad Software, San Diego, CA, USA). Values in tables and graphs are expressed as mean ± SEM. Two groups of mice were compared with Student's two-tailed t-test for independent data and $p > 0.05$ was considered not significant (ns); $*p < 0.05$, $**p < 0.01$, $***p < 0.001$, and $****p < 0.0001$ were considered significant.

## Acknowledgements

We are grateful to all the participants for their cooperation. We appreciated Li Wang and Dandan Song at the Center of Cryo-Electron Microscopy, Zhejiang University, for their technical assistance with TEM and SEM, respectively. We thank the Bioinformatics Center of the USTC, School of Life Sciences, for providing supercomputing resources. This work was supported by the National Key Research and Development Program of China (2022YFC2702601, 2022YFA0806303, and 2019YFA0802600), the National Natural Science Foundation of China (82171599, 32270901, and 32070850), the Global Select Project (DJK-LX-2022010) of the Institute of Health and Medicine, Hefei Comprehensive National Science Center, the Joint Fund for New Medicine of USTC (YD9100002034) and Scientific Research Foundation for Scholars of the First Affiliated Hospital of USTC (RC2023054).

# Additional information

## Funding

| Funder | Grant reference number | Author |
| --- | --- | --- |
| National Key Research and Development Program of China | 2022YFC2702601 | Qinghua Shi |
| National Key Research and Development Program of China | 2022YFA0806303 | Qinghua Shi |
| National Key Research and Development Program of China | 2019YFA0802600 | Qinghua Shi |
| Natural Science Foundation of China | 82171599 | Qinghua Shi |
| Natural Science Foundation of China | 32270901 | Qinghua Shi |
| Natural Science Foundation of China | 32070850 | Qinghua Shi |
| Institute of Health and Medicine | DJK-LX-2022010 | Qinghua Shi |
| Hefei Comprehensive National Science Center, the Joint Fund for New Medicine of USTC | YD9100002034 | Qinghua Shi |
| Scientific Research Foundation for Scholars of the First Affiliated Hospital of USTC | RC2023054 | Baolu Shi |

The funders had no role in study design, data collection and interpretation, or the decision to submit the work for publication.

## Author contributions

Menglei Yang, Data curation, Formal analysis, Validation, Investigation, Visualization, Methodology, Writing – original draft, Writing – review and editing; Hafiz Muhammad Jafar Hussain, Manan Khan, Zubair Muhammad, Ao Ma, Xiongheng Huang, Jingwei Ye, Min Chen, Aoran Zhi, Tao Liu, Ranjha Khan, Aurang Zeb, Nisar Ahmad, Investigation, Methodology; Jianteng Zhou, Huan Zhang, Resources, Software; Ali Asim, Wasim Shah, Methodology; Bo Xu, Supervision; Hui Ma, Supervision, Writing – review and editing; Qinghua Shi, Conceptualization, Resources, Funding acquisition, Writing – review and editing; Baolu Shi, Conceptualization, Funding acquisition, Writing – review and editing

## Author ORCIDs

Baolu Shi ⓘ https://orcid.org/0000-0002-5791-4300

## Ethics

This work was approved by the institutional ethics committee of the University of Science and Technology of China (USTC) with the approval number 2019-KY-168, and informed consent was obtained from patients and controls.

All animal experiments were approved by the Institutional Animal Care and Use Committee of the USTC with the approval number USTCACUC1301021.

Reviewer #1 (Public review): https://doi.org/10.7554/eLife.100350.3.sa1
Reviewer #2 (Public review): https://doi.org/10.7554/eLife.100350.3.sa2
Author response https://doi.org/10.7554/eLife.100350.3.sa3

## Additional files

### Supplementary files

Supplementary file 1. The used reagents and protein interactors information. (a) List of candidate dynein protein interactors of DNAH12. (b) Primers used in the study. Primer used for Sanger Sequencing of the identified mutations, genotyping of the mice models, Reverse Transcription Polymerase Chain Reaction (RT-PCR) and Quantitative Polymerase Chain Reaction (qPCR). (c) The primary antibodies used in this study. (d) The secondary antibodies used in this study. (e) Primers used for the construction of plasmids in co-immunoprecipitation (Co-IP assays).

MDAR checklist

### Data availability

The proteomics data are available on the ProteomeXchange Consortium via the iProX partner repository (*Chen et al., 2022*), with the dataset identifier PXD051681.

The following dataset was generated:

| Author(s) | Year | Dataset title | Dataset URL | Database and Identifier |
|---|---|---|---|---|
| Yang M | 2024 | Biallelic mutations in DNAH12 cause asthenoteratozoospermia in humans and mice | https://proteomecentral.proteomexchange.org/cgi/GetDataset?ID=PXD051681-1 | ProteomeXchange, PXD051681 |

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
