## [Editor Report · eLife Assessment]

This **fundamental** study further validates DNAH12 as a causative gene for asthenoteratozoospermia and male infertility in both humans and mice. **Compelling** evidence supports the notion that DNAH12 is essential for proper axonemal development. This work will be of interest to reproductive biologists studying spermatogenesis and sperm biology, as well as andrologists focusing on male fertility.

---

## [Referee Report · Reviewer #1 (Public review)]

Summary:

Even though this is not the first report that the mutation in the DNAH12 gene causes asthenoteratozoospermia, the current study explores the sperm phenotype in-depth. The authors show experimentally that the said mutation disrupts the proper axonemal arrangement and recruitment of DNALI1 and DNAH1 - proteins of inner dynein arms. Based on these results, the authors propose a functional model of DNAH12 in proper axonemal development. Lastly, the authors demonstrate that the male infertility caused by the studies mutation can be rescued by ICSI treatment at least in the mouse. This study furthers our understanding of male infertility caused by a mutation of axonemal protein DNAH12, and how this type of infertility can be overcome using assisted reproductive therapy.

Strengths:

This is an in-depth functional study, employing multiple, complementary methodologies to support the proposed working model.

Weaknesses:

The structure and interaction model between DNAH12, DNALI1, and DNAH1 relies on in silico methodologies, and further studies are required to validate these predictions.

---

## [Referee Report · Reviewer #2 (Public review)]

Summary:

The authors first conducted whole exome sequencing for infertile male patients and families where they co-segregated the biallelic mutations in the Dynein Axonemal Heavy Chain 12 (DNAH12) gene. Sperm from patients with biallelic DNAH12 mutations exhibited a wide range of morphological abnormalities in both tails and heads, reminiscing a prevalent cause of male infertility, asthenoteratozoospermia. To deepen the mechanistic understanding of DNAH12 in axonemal assembly, the authors generated two distinct DNAH12 knockout mouse lines via CRISPR/Cas9, both of which showed more severe phenotypes than observed in patients. Ultrastructural observations and biochemical studies revealed the requirement of DNAH12 in recruiting other axonemal proteins and that the lack of DNAH12 leads to the aberrant stretching in the manchette structure as early as stage XI-XII. At last, the authors proposed intracytoplasmic sperm injection as a potential measure to rescue patients with DNAH12 mutations, where the knockout sperm culminated in the blastocyst formation with a comparable ratio to that in WT.

Strengths:

The authors convincingly showed the importance of DNAH12 in assembling cilia and flagella in both human and mouse sperm. This study is not a mere enumeration of the phenotypes, but a strong substantiation of DNAH12's essentiality in spermiogenesis, especially in axonemal assembly.

The analyses conducted include basic sperm characterizations (concentration, motility), detailed morphological observations in both testes and sperm (electron microscopy, immunostaining, histology), and biochemical studies (co-immunoprecipitation, mass-spec, computational prediction). Molecular characterizations employing knockout animals and recombinant proteins beautifully proved the interactions with other axonemal proteins.

Many proteins participate in properly organizing flagella, but the exact understanding of the coordination is still far from conclusive. The present study gives the starting point to untangle the direct relationships and order of manifestation of those players underpinning spermatogenesis. Furthermore, comparing flagella and trachea provides a unique perspective that attracts evolutional perspectives.

Weaknesses:

Seemingly minor, but the discrepancies found in patients and genetically modified animals were not fully explained. For example, both knockout mice vastly reduced the count of sperm in the epididymis and the motility, while phenotypes in patients were rather milder. Addressing the differences in the roles that the orthologs play in spermatogenesis would deepen the comprehensive understanding of axonemal assembly.

Comments on revisions:

The reviewer is satisfied with the authors' response.

---

## [Author Response]

The following is the authors’ response to the original reviews.

eLife AssessmentThis important study further validates DNAH12 as a causative gene for asthenoteratozoospermia and male infertility in humans and mice. The data supporting the notion that DNAH12 is required for proper axonemal development are generally convincing, although more experiments would solidify the conclusions. This work will interest reproductive biologists working on spermatogenesis and sperm biology, as well as andrologists working on male fertility.

We thank the editor and the two reviewers for their time and careful evaluation of our manuscript. We sincerely appreciate their encouraging feedback and insightful guidance on improving our study. In the revised manuscript, we have performed additional experiments and provided quantitative data regarding the reviewers' comments.

**Public Reviews:**

**Reviewer #1 (Public Review):**
Summary:Even though this is not the first report that the mutation in the DNAH12 gene causes asthenoteratozoospermia, the current study explores the sperm phenotype in-depth. The authors show experimentally that the said mutation disrupts the proper axonemal arrangement and recruitment of DNALI1 and DNAH1 - proteins of inner dynein arms. Based on these results, the authors propose a functional model of DNAH12 in proper axonemal development. Lastly, the authors demonstrate that the male infertility caused by the studies mutation can be rescued by ICSI treatment at least in the mouse. This study furthers our understanding of male infertility caused by a mutation of axonemal protein DNAH12, and how this type of infertility can be overcome using assisted reproductive therapy.Strengths:This is an in-depth functional study, employing multiple, complementary methodologies to support the proposed working model.

Thank you for your recognition of the strength of this study. Your positive feedback motivates us to continue refining our research and methodological rigor in future studies.

Weaknesses:The study strength could be increased by including more controls such as peptide blocking of the inhouse raised mouse and rat DNAH12 antibodies, and mass spectrometry of control IP with beads/IgG only to exclude non-specific binding. Objective quantifications of immunofluorescence images and WB seem to be missing. At least three technical replicates of western blotting of sperm and testis extracts could have been performed to demonstrate that the decrease of the signal intensity between WT and mutant was not caused by a methodological artifact.

Thank you for your comments. In order to study in-depth, we have analyzed the protein sequence features of DNAH12 protein, 1-200 amino acids of DNAH12 were selected as the ideal antigen considering its good performance (1. high immunogenicity; 2. High hydrophilicity; 3. Good Surface Leakage Groups; 4. Sequence homology analysis to avoid unspecific recognition to other proteins;). The two different anti-DNAH12 antibodies were developed with the help Dia-An Biotech company in 2022, we have tried to acquire the polypeptide fragments of target proteins to do peptide blocking but the material were discard after the service. Luckily, we have got the target band of DNAH12 protein in western blotting experiment while the band was not detected in knockout mice group; the immunofluorescence signals of DNAH12 were strong but not present in knockout mice group. Besides, we have tested that the inhouse raised rabbit antibody were suitable for IP experiment. The IP experiment also showed the raised rabbit antibody were able to immunoprecipitated the DNAH12 band in the *Dnah12*^+/+^ mice but not in *Dnah12*^-/-^ mice. Collectively, these data could support the specificity of the raised DNAH12 antibodies. In IP assay, we have added the IgG group in the IP-mass spectrometry to exclude non-specific binding. And the experimental design was described in Figure 6B. The raw data were deposited in iProX partner repository (accession number: PXD051681), and we have coordinated with the repository manager to make the data publicly accessible (https://www.iprox.cn/page/subproject.html?id=IPX0008674001).

Besides, we have conducted replicates of western blotting of sperm and testis extracts at least 3 times and added the objective quantifications of immunofluorescence signals and WB images. The quantifications of the blot were shown in figures to help readers understand these results easily.

**Reviewer #2 (Public Review):**
Summary:The authors first conducted whole exome sequencing for infertile male patients and families where they co-segregated the biallelic mutations in the Dynein Axonemal Heavy Chain 12 (DNAH12) gene.Sperm from patients with biallelic DNAH12 mutations exhibited a wide range of morphological abnormalities in both tails and heads, reminiscing a prevalent cause of male infertility, asthenoteratozoospermia. To deepen the mechanistic understanding of DNAH12 in axonemal assembly, the authors generated two distinct DNAH12 knockout mouse lines via CRISPR/Cas9, both of which showed more severe phenotypes than observed in patients. Ultrastructural observations and biochemical studies revealed the requirement of DNAH12 in recruiting other axonemal proteins and that the lack of DNAH12 leads to the aberrant stretching in the manchette structure as early as stage XI-XII. At last, the authors proposed intracytoplasmic sperm injection as a potential measure to rescue patients with DNAH12 mutations, where the knockout sperm culminated in the blastocyst formation with a comparable ratio to that in WT.Strengths:The authors convincingly showed the importance of DNAH12 in assembling cilia and flagella in both human and mouse sperm. This study is not a mere enumeration of the phenotypes, but a strong substantiation of DNAH12's essentiality in spermiogenesis, especially in axonemal assembly.The analyses conducted include basic sperm characterizations (concentration, motility), detailed morphological observations in both testes and sperm (electron microscopy, immunostaining, histology), and biochemical studies (co-immunoprecipitation, mass-spec, computational prediction). Molecular characterizations employing knockout animals and recombinant proteins beautifully proved the interactions with other axonemal proteins.Many proteins participate in properly organizing flagella, but the exact understanding of the coordination is still far from conclusive. The present study gives the starting point to untangle the direct relationships and order of manifestation of those players underpinning spermatogenesis. Furthermore, comparing flagella and trachea provides a unique perspective that attracts evolutional perspectives.

Thank you for your thoughtful and positive feedback. We are delighted that you found our study to be a strong substantiation of DNAH12's essential role in spermiogenesis, particularly in axonemal assembly. We believe that this study represents a meaningful step toward unraveling the intricate coordination of axonemal proteins during spermatogenesis, and your comments further inspire us to continue exploring these complex mechanisms in future work. Thank you once again for your valuable insights and summary of this work.

Weaknesses:Seemingly minor, but the discrepancies found in patients and genetically modified animals were not fully explained. For example, both knockout mice vastly reduced the count of sperm in the epididymis and the motility, while phenotypes in patients were rather milder. Addressing the differences in the roles that the orthologs play in spermatogenesis would deepen the comprehensive understanding of axonemal assembly.

This is an interesting question. Actually, it seems that although humans and mice share the male infertility phenotypes with deficiency in dynein proteins essential for sperm flagellar development, they are different in some ways. For instance, it has been reported that deficiency in DNAH17 (Clin Genet. 2021. PMID: 33070343) or DNAH8 (Am J Hum Genet. 2020. PMID: 32619401; PMCID: PMC7413861), two other members of Dynein Axonemal Heavy Chain family, also cause more severe phenotype in mice, comparing with that of human patients carrying bi-allelic DNAH17 or DNAH8 loss-of-function mutations. In knockout mice, sperm counts are lower, and the proportion of abnormal sperm morphology is higher, whereas the phenotypes in human patients tend to be milder. These observations suggest that orthologs may influence spermatogenesis to slightly different extents in humans and mice. We plan to investigate the mechanisms underlying these discrepancies in future studies, which will provide deeper insights into axonemal assembly and the evolutionary aspects of spermatogenesis. Thank you again for bringing up this important issue.

**Recommendations for the authors:**

**Reviewer #1 (Recommendations For The Authors):**
This reviewer is impressed by the study's depth and the extent of the methodology used in the study. The study is well-designed, and the results are very interesting. The reviewer's enthusiasm was reduced by the lack of some controls (provided that the reviewer did not miss them). Further are point-to-point suggestions that this reviewer believes will increase the merit of the present study.Title:(1) Why a "special" dynein? What makes it special when compared to other dyneins? I suggest removing the word special.

Through phylogenetic and protein domain analyses of the DNAH family, we found that DNAH12 is the shortest member and the only one that lacks a typical microtubule-binding domain (MTBD) in the DNAH family, thus we want to describe it as a “special” dynein. We have fully considered your valuable suggestion and decided to remove it from the title.

Abstract:(2) L23: same as above, why special?

We identified DNAH12 as the shortest member of the DNAH family and uniquely lacking the typical microtubule-binding domain (MTBD). This distinct characteristic prompted us to describe it as a 'special' dynein in the abstract part.

(3) L37: the reviewer did not find a figure (neither main nor supplementary) that would demonstrate the proper organization of microtubules in cilia. Figure S11 only shows the presence of cilia in DNAH12-/- mouse. A TEM image of cilia is required to confirm or reject the claim that DNAH12 does not play a crucial role in proper microtubule organization in cilia.

We have now added TEM images of cilia in wild-type and *Dnah12*^-/-^ mice. The ultra-structures of cilia axonemes were comparable in wild-type and *Dnah12*^-/-^ groups, suggesting that DNAH12 may not play crucial role in proper microtubule organization. The results have now been added to Supplemental Figure 11F.

(4) L122-6: Did the authors also confirm these structures by cryo-EM? If not, this needs to be pointed out as a shortcoming in the discussion, that the structures and interactions are predicted in silico only.

Thank you for your comment. Due to resource limit, we do not perform cryo-EM to confirm these structures. We will pursue the structures details at an atomic resolution structure in further study. We understand this point and now we have addressed this as a shortcoming in the discussion part.

(5) L134: Be more specific about what characteristics of DNAH12 were analyzed.

Thank you for your comment. We have now updated these in the method part. The characteristics of the DNAH12 including its region immunogenicity, hydrophilicity, surface leakage groups, and sequence homology were analyzed.

(6) L137: Be more specific about how the antibodies validated were. Were the antibodies validated for both immunofluorescence and western blotting? I suggest doing peptide blocking of the antibody, for instance for ICC, preincubation of ab with immunizing peptide followed by primary ab incubation with studied cells/tissues.

Thank you for your comments and suggestions. We validated the antibodies for both immunofluorescence and western blotting to ensure their effectiveness in our experiments. The two different anti-DNAH12 antibodies were developed with the help Dia-An Biotech company in 2022, we have attempted to acquire the polypeptide fragments of target proteins to do peptide blocking but the material were disposed after the service. Luckily, we have got the target band of DNAH12 protein which showed strong signal in western blotting experiment and the band was not detected in knockout mice group; the immunofluorescence signals of DNAH12 were strong but not present in knockout mice group. Besides, the IP experiment also showed the raised rabbit antibody were able to immunoprecipitated the DNAH12 band in the *Dnah12*^+/+^ mice but not in *Dnah12*^-/-^ mice. Collectively, these data could support the specificity of the raised DNAH12 antibodies. We sincerely admire your suggestion and will require for the peptide material if we develop new antibodies.

(7) L142: This reviewer is unfamiliar with using TRIzol for sperm protein extraction. Is there a specific reason for not using PAGE loading buffer for human sperm protein extraction?

Thanks for your suggestions. TRIzol reagent can be used for small amounts of samples (5×10^6^ cells) as well as large amounts of samples (>10^7^ cells). It is suitable for extraction of RNA and proteins at the same time. Our lab has adopted these methods in our previous work (Hum Reprod Open. 2023; PMID: 37325547; PMCID: PMC10266965.). This method is very useful to process valuable small amounts of samples for scientific work. The human sperm protein extraction was added with SDS-sample buffer [PAGE loading buffer] before SDS-PAGE separation. We have added this detail in the method part. We are sorry for making this misunderstanding.

(8) L144: Were these the final concentrations of the SDS loading buffer? 1 × Laemmli buffer contains 62.5 mM TRIS, 2% (w/w) SDS, 10 % (w/v) glycerol, and 5% 2-mercaptoethanol. Please, amend accordingly.

Thanks for your suggestions. We apologized for incorrect labelling of concentrations (The previous one is 3× SDS loading buffer). We have now amended the SDS loading buffer to 1 × Laemmli buffer as suggested.

(9) L151: Table S2 contains other homemade antibodies than DNAH12. Please, include references to the studies where the generation and validation of these antibodies is described.

Thank you for your suggestions. We have developed a DNAH1 antibody for use in Western blot assays, with its generation and validation detailed in Frontiers in Endocrinology (Lausanne), 2021 (PMID: 34867808; PMCID: PMC8635859). Additionally, we have produced a DNAH17 antibody for both immunofluorescence (IF) and Western blot, as described in Journal of Experimental Medicine, 2020 (PMID: 31658987; PMCID: PMC7041708). These references have now been included.

(10) L167: Please, spell out ICR at its first appearance.

Done as suggested, Thank you. The full name of ICR is Institute of Cancer Research.

(11)L169: This reviewer is confused. It seems that the mouse encodes DNAH12 on exons 5 and 18 simultaneously. Each mouse model has only one exon targeted for a knockout. Would not this mean that the expression of DNAH12 in both models is not completely knocked down? Please, give more background in this paragraph for those less familiar with CRISPR/Cas9.

Thank you for your insightful comment. We appreciate your attention to detail. To clarify, while the mouse model does indeed encode DNAH12 on exons 5 and 18 simultaneously, we specifically targeted the key exon 5 or exon 18 in each model to achieve different knockout strategies. This approach allows us to assess the functional implications of the remaining DNAH12 expression in both models. We have checked the DNAH12 expression in both models, and the result showed both models present with undetected DNAH12 proteins, indicating both models were completely knocked out of DNAH12 proteins. Additionally, we will revise the manuscript to include further details on the CRISPR/Cas9 methodology, ensuring accessibility for readers less familiar with this technique. Thank you again for your valuable feedback, which we believe will greatly enhance our manuscript.

(12) L201: 50 % PBS? As in 0.5 x concentrated PBS? Please, rewrite for clarity.

The term "50% PBS" refers to a 1:1 dilution of phosphate-buffered saline (PBS) with an appropriate diluent, resulting in a final concentration of 0.5x PBS. We will revise the text to explicitly clarify this, ensuring it is clear to all readers. Thank you for highlighting this point.

(13) L224: Please, state what beads those were (magnetic/agarose, conjugated to protein A/G...) Include catalog # and manufacturer.

Thank you for your suggestion. We have updated the manuscript to include this information. The beads used were Protein A/G Magnetic Beads (Catalog #B23202, Bimake, Texas, USA).

(14) L227: What was the reason for adding a proteasomal inhibitor? What concentration was used? Please, add this information to the text.

We adding MG132 in cell immunoprecipitation (IP) experiments is to inhibit proteasomal activity, thereby preventing the degradation of the target protein. This helps maintain the stability of the target protein during the experiment (Sci Adv. 2022. PMID: 35020426; PMCID: PMC8754306.), enhancing its detectability in subsequent analyses. MG132 (5 μM) was added. We have added this information in the revised the manuscript

(15) L233: in vivo IP of mouse testis lysate? This does not make sense. I suggest removing "in vivo".

Thank you for your careful review and comments on our manuscript. We have modified as suggested.

(16) L317: Supplemental Figure 6 precedes Supplemental Figure 5 in the text, which is neither logical nor orderly.

Thank you for your suggestion. Since the N-terminal DNAH12 antibody is already described in the Methods section (L317), we propose removing Supplemental Figure 6 from the content to improve the logical flow and maintain an orderly presentation.

(17) L345 and elsewhere: how did the authors quantify the decrement of the signal? This needs to be measured objectively.

Thank you for your valuable suggestion. We quantified the signal intensity using Fiji (Nat Methods. 2012. PMID: 22743772; PMCID: PMC3855844), which allows for precise analysis of pixel intensity. The results are presented in the figures to effectively illustrate the decrement in signal intensity. We appreciate your suggestion, and we have provided a description of the method in our methodology section.

(18) L371: I recommend: ...and elongated spermatids; the abnormal...

Done as suggested. Thank you.

(19) L412-4: Cilia in both *Dnah12*^mut/mut^ and *Dnah12*^-/-^ are developed, but are they motile or immotile? This needs to be investigated. Is the DNAH12 in cilia truncated while still fulfilling its function?

Thanks for your comment. We have checked the ciliary motility using an inverted microscope, and no significant difference of ciliary motility were observed between the knockout group and the control group. These results indicated that the ciliary motility was not affected by DNAH12 deficiency. The N-terminal DNAH12 antibody was developed to detect whether a truncated protein in mice tissues while we do not detect DNAH12 signals through immunofluorescence assay on trachea sections of the *Dnah12*^-/-^ mice. These results indicate that DNAH12 may exert little influence on cilia, comparing to its important function in flagella.

(20) L414-6: The results do not support this claim as the authors do not show that cilia are motile.

Thanks for your comment. The supplemental videos 3-4 of trachea live of *Dnah12*^+/+^ and *Dnah12*^-/-^ mice have been uploaded to support this conclusion.

(21) L421-3: Did the authors perform a negative test, where they let the testis lysate interact with beads/IgG only and performed the MS to identify non-specific binding? This is a crucial specificity test for this approach.

We have performed negative test. In IP assay, we have added the IgG group in the IP-mass spectrometry to exclude non-specific binding. And the experimental design was described in Figure 6B. The raw data were deposited in iProX partner repository (PXD051681), which we have required the manager soon to update the status to public, so it will be visible to readers.

(22) L462: same as #18 the authors need to show that cilia are also motile. The mere presence of cilia in DNAH12-/- as shown in Fig S11C&D is not sufficient to conclude that the mice do not manifest PCD symptoms.

Thanks for your comment. We do not observe obvious differences between the cilia of *Dnah12*^+/+^ and *Dnah12*^-/-^ mice. The supplemental videos 3-4 of trachea live of *Dnah12*^+/+^ and *Dnah12*^-/-^ mice have been uploaded to show the motility of the trachea.

(23) L529: MTBD region instead of domain, as "domain" is already part of the abbreviation.

Done as suggested

(24) L875: Sperm is both the singular and plural form. Spermatozoon vs spermatozoa can be used where the distinction between singular and plural needs to be made.

Thanks for your suggestion. We have checked and changed this usage.

(25) Figure 3H: Is there a specific reason why P11 is not shown?

Because limited smear slides of P11 were available, the P11 were not stained for DNAH17 antibody previously. We have now updated the experiment, which showed that DNAH17 expression were not affected in patient P11. We have now added this result to Figure 3H.

(26) Figure 8H: The authors in their MS do not describe what is happening to N-DRC proteins, yet they suggest in their model that it's unaffected in the mutant mouse/human. Please, address this in the MS and clearly state in the model that N-DRC needs further exploration in future studies.

Thanks for your suggestion, we have checked the MS data but do not observe the enrichment of nexin-dynein regulatory complex (N-DRC) protein, just one known N-DRC protein DRC1 present with only 1 unique peptide. Instead, enrichment of inner dynein arm proteins and radial spoke proteins were observed. However, we cannot determine the N-DRC structures maybe affected or not. We have stated this in the discussion part and will pursue this with high resolution technology like cryo-EM in the future.

(27) Figure 5F: Is it possible to choose a different *Dnah12*^-/-^ spermatozoon to see a reduced level of DNALI1 so that it corresponds with the WB detection in Fig 5B?

Thanks for your suggestion, we have chosen a *Dnah12*^-/-^ spermatozoon with faint remnants of the DNALI1 signal as the representative picture.

(28) Figure S2 and elsewhere: How were the authors able to resolve and calibrate 356 kDa protein using SDS PAGE? Agarose electrophoresis protein electrophoresis is more suitable for resolution of high molecular proteins. Most of the protein standards have as high molecular standard as 250 kDa.

We have found that high molecular proteins (like 356kDa) were able to resolve in concentration 4-12% gradient gel of polyacrylamide gels and employ appropriate voltages and more time during electrophoresis to improve resolution of high molecular weight proteins. The DNAH12 proteins were calibrated by the using of a HiMark Pre-Stained High Molecular Weight Protein Standard (30-460 kDa). We have now updated the blot images to show the size of the DNAH12 protein (Fig S6B,). The target band is obvious between 268 kDa and 460 kDa, which make it easy to calculate the target band of DNAH12 antibody elsewhere. Thanks for your suggestion.

(29) Figure S5: similar to #24: Why P10 and P11 are not shown?

Because limited smear slides of P10 or P11 were available, we did not stain ODF2 antibody previously. We have now updated the experiments, which showed that ODF2 expression were not affected in patient P10 or P11. We have now added this result to Figure S5.

(30) Figure S6B: The specificity of the anti-DNAH12 antibody against mouse DNAH12 seems to be questionable since the authors detect multiple bands on WB. I recommend doing peptide blocking to show that these are non-specific binding as opposed to off-target binding.

Thank you for your comments. In order to study in-depth, we have analyzed the protein sequence features of DNAH12 protein, 1-200 amino acids of DNAH12 were selected as the ideal antigen considering its good performance (1. high immunogenicity; 2. High hydrophilicity; 3. Good Surface Leakage Groups; 4. Sequence homology analysis to avoid unspecific recognition to other proteins;). The two different anti-DNAH12 antibodies were developed with the help Dia-An Biotech company in 2022, we have attempted to acquire the polypeptide fragments of target proteins to do peptide blocking but the material were disposed after the service. Luckily, we have got the target band of DNAH12 protein which showed strong signal in western blotting experiment and the band was not detected in knockout mice group; the immunofluorescence signals of DNAH12 were strong but not present in knockout mice group. Besides, we have tested that the inhouse raised rabbit antibody was suitable for IP experiment. The IP experiment also showed the raised rabbit antibody were able to immunoprecipitated the DNAH12 band in the *Dnah12*^+/+^ mice but not in *Dnah12*^-/-^ mice. Collectively, these data could support the specificity of the raised DNAH12 antibodies. We admire your suggestion and will require for the peptide material if we develop new antibodies.1

**Reviewer #2 (Recommendations For The Authors):**
Recruitment of DNAH1 and DNALI1 to the flagella is dependent on DNAH12 expression, according to the data. What would be the mechanism that locates DNAH12 which lacks MTBD to the flagella?

Thank you for your insightful question. We are currently investigating the mechanisms that facilitate the loading of DNAH12 to the flagella. Based on existing data, we hypothesize that CCDC39 and/or CCDC40 may play a critical role in the recruitment of DNAH12 to sperm flagella during spermiogenesis (Nat Genet. 2011, PMID: 21131972; PMCID: PMC3509786; Nat Genet. 2011, PMID: 21131974; PMCID: PMC3132183). Furthermore, a structural study by Walton et al. showed that DNAH12 associates with CCDC39/CCDC40 proteins (Nature. 2023, PMID: 37258679; PMCID: PMC10266980). These findings suggest that CCDC39 and/or CCDC40 may play a role in facilitating the localization of DNAH12 to the flagella. Additional studies are needed to identify other potential factors involved in this process and to further elucidate the mechanisms underlying this complex biological phenomenon.